# Stochastic Principal-Agent Problems: Computing and Learning Optimal History-Dependent Policies

**Jiarui Gan**
University of Oxford
jiarui.gan@cs.ox.ac.uk

**Rupak Majumdar**
MPI-SWS
rupak@mpi-sws.org

**Debmalya Mandal**
University of Warwick
debmalya.mandal@warwick.ac.uk

**Goran Radanovic**
MPI-SWS
gradanovic@mpi-sws.org

## Abstract

We study a stochastic principal-agent model. A principal and an agent interact in a stochastic environment, each privy to observations about the state not available to the other. The principal has the power of commitment, both to elicit information from the agent and to signal her own information. The players communicate with each other and then select actions independently. Both players are *far-sighted*, aiming to maximize their total payoffs over the entire time horizon. We consider both the computation and learning of the principal's optimal policy. The key challenge lies in enabling *history-dependent* policies, which are essential for achieving optimality in this model but difficult to cope with because of the exponential growth of possible histories as the size of the model increases; explicit representation of history-dependent policies is infeasible as a result. To address this challenge, we develop algorithmic techniques based on the concept of *inducible value set*. The techniques yield an efficient algorithm that computes an $\epsilon$-approximate optimal policy in time polynomial in $1/\epsilon$. We also present an efficient learning algorithm for an episodic reinforcement learning setting with unknown transition probabilities. The algorithm achieves sublinear regret $\widetilde{\mathcal{O}}(T^{2/3})$ for both players over $T$ episodes.

## 1 Introduction

Many problems in economic theory involve sequential reasoning between multiple parties with asymmetric access to information [37, 20, 4, 27]. For example, in contract theory, one party (the principal) delegates authority and decision-making power to another (the agent), and the goal is to design mechanisms to ensure that the agent's actions align with the principal's utilities. This broad class of *principal-agent problems* lead to many research questions about information design and optimal strategic behaviors, with broad-ranging applications from governance and public administration to e-commerce and financial services. In particular, *algorithmic* techniques for optimal decision-making and learning are crucial for obtaining effective solutions to real-world problems in this domain.

In this paper, we consider a general framework for stochastic principal-agent problems. We study the algorithmic problems related to the computation and learning of optimal solutions under this framework. In this framework, the interaction between the principal and the agent takes place in a stochastic environment over multiple time steps. In each step, both players are privy to information not available to the other and make partial observations about the environment. The players can communicate their private information to influence each other and, based on this communication, play actions that jointly influence the state of the environment. Each player has their own payoff, and we study the *general-sum case* with *far-sighted* players: each player aims to maximize the (expected)

39th Conference on Neural Information Processing Systems (NeurIPS 2025).

sum of their rewards over the entire horizon of the game. Technically, the setting is a stochastic game with partial information on both sides [1, 31].

In line with the principal-agent framework, we assume that the principal has the power of *commitment*, both to elicit information from the agent and to signal the agent about her own private information. A commitment is a binding agreement for the principal to act according to the committed policy. The agent acts optimally in response to the commitment, deciding what information to share and what actions to perform at their own discretion. At a high level, we have a Stackelberg game and we aim to find a Stackelberg equilibrium [40]. The possibility of information exchange further results in a model that incorporates both sequential Bayesian persuasion (or information design [22]) [16, 41] and sequential mechanism design [45]. We adopt *stochastic games* (or Markov games) [39] as our modeling basis, whose transition models are graph-structured. Hence, our model is more expressive than tree-structured principal-agent problems such as those based on extensive-form games (EFGs), in the sense that exponentially many possible history trajectories can be unrolled from our graph-based model as the length of the time horizon increases, whereas in tree-structured ones, the number is bounded by the size of the tree.

Indeed, *history-dependence* is a key consideration in our policy design. Unlike in single-player Markov decision processes (MDPs) or zero-sum stochastic games, stationary (or history-*in*dependent) policies are no longer without loss of generality in our setting. For example, strategies like tit-for-tat—which are essential for achieving optimal outcomes in certain scenarios (such as the repeated prisoner's dilemma)—depend on remembering past actions and thus cannot be implemented by stationary policies. Enabling history-dependent policies in our graph-based model, however, begets a major challenge caused by the exponential growth of possible histories. Explicitly representing a history-dependent policy requires specifying a strategy for every possible history, which quickly becomes infeasible as the number of histories increases exponentially with the length of the time horizon. To further optimize such a policy is even more challenging as the output of this computational task would be a function over an exponentially large domain. In contrast, the size of a stationary policy is linear in the number of states.

## 1.1 Our Contributions

Surprisingly though, as we discover in this work, history-dependent policies can be represented as value vectors and can be efficiently unrolled provided polytopes consisting of all inducible value vectors. Building on this insight, we develop algorithmic techniques that work by constructing the *inducible value sets*. These techniques make it possible to implement and optimize history-dependent strategies in dynamic principal-agent settings. More surprisingly, they are computationally efficient, so not only do they significantly expand the policy space—from stationary policies to the far more expressive class of history-dependent ones—but they also address a key computational barrier posed by stationary policies, to optimize which, even approximately, is known to be NP-hard [15].

In summary, we make the following contributions in this paper.

- We introduce a generalized framework for principal-agent problems in stochastic environments, which incorporates the stochastic extensions of various subcases of principal-agent problems such as information design [22] and automated mechanism design [38].

- We develop algorithmic techniques that enable the implementation and optimization of history-dependent strategies in the stochastic principal-agency framework. The techniques are based on a compact representation of fully history-dependent polices and a novel approach that works by constructing the inducible value sets. Together, they yield an efficient near-optimal algorithm that computes an $\epsilon$-approximate optimal policy in time polynomial in $1/\epsilon$, while the policy it produces ensures *exact* incentive compatibility (IC).

- We also consider an episodic RL setting where the transition dynamics are unknown and must be learned through interaction. We present a learning algorithm that guarantees sublinear regret $\widetilde{\mathcal{O}}(\text{poly}(M, H) \cdot T^{2/3})$ for both players, where $M$ is the model size, $H$ is the episode length, and $T$ is the number of episodes. The algorithm builds on recent advances in *reward-free exploration* from the RL literature and it leverages efficient computation of *approximately* IC policies, using a variant of our algorithm for the full-information setting.

## 1.2 Related Work

The principal-agent problem is a well-known concept in economics studies [see, e.g., 37, 33, 32, 29]. Models featuring sequential interactions have also been proposed and studied [34, 12]. Our work follows the same modeling approach as these early works and extends the one-shot versions of the respective models—such as information design [22], automated mechanism design [38], as well as mixtures of the two [33, 7, 14]—into the sequential setting. There has been a growing interest in the algorithmic aspects of these sequential models, focusing on their computation and learning (e.g., information design [8, 15, 16, 41, 2], automated mechanism design [45, 6], other types of sequential Stackelberg games [25, 26, 5, 18, 10], and even more recently, contract design [42, 19]).

More specifically, Gan et al. [15] first introduced an infinite-horizon information design model based on an MDP. They showed that optimal *stationary* strategies are inapproximable, unless the receiver is myopic. This work left open the tractability of optimal *history-dependent* strategies, especially in finite-horizon models, which we consider in this paper. Wu et al. [41] later studied the reinforcement learning problem against a myopic agent in the same sequential information design model. Bernasconi et al. [2] also studied the same problem in a model based on an EFG; efficient computation and learning algorithms were presented. Similar EFG-based models have also been explored in the recent literature [43, 44]. EFGs are tree-structured and hence easier than MDP-based models in the sense that all history sequences are modeled explicitly in the model. The number of possible histories is bounded by the size of the problem as a result, where as this can be exponential in an MDP. Hence, efficient algorithms for EFG-based models do not directly translate to efficient algorithms for our MDP-based model. In the domain of automated mechanism design, Zhang and Conitzer [45] studied a finite-horizon model that is a POMDP for the principal and MDP for the agent. They presented a linear program for computing optimal mechanisms, whose size is exponential in the problem size.

As mentioned earlier, our algorithm leverages compact representation of history-dependent polices and construction of inducible value sets. Similar techniques have been proposed in earlier works by Dermed and Isbell [11] and MacDermed et al. [28] to compute optimal correlated equilibria of stochastic games. Our algorithm extends these techniques into the principal-agent setting, with adaptions that ensure exact incentive compatibility (IC) in the full-information setting. Concurrently with our work [17], Bernasconi et al. [3] also used a similar approximation approach to solve an information design problem (as a special case of our model). Compared to their results, our algorithm guarantees *exact* IC, with a more straightforward approach. Moreover, we also study the learning problem where the transition model needs to be learned, in addition to the computation problem where this is given. While all the above works (including ours) only guarantee near-optimality, exact solutions are possible in some special cases. In a recent work of Zhang et al. [46], they presented a sophisticated exact algorithm for computing optimal correlated equilibria in two-player turn-based stochastic games. In infinite-horizon games, however, such exact solutions appear computationally infeasible because of irrational numbers involved in optimal polices; we refer the reader to a more recent work by Gan and Majumdar [13].

## 2 Preliminaries

A principal (P) and an agent (A) interact in a finite-horizon POMDP $\mathcal{M} = \langle S, A, \Omega, p, \mathbf{r} \rangle$, where: $S$ is a finite state space; $A = A^{\mathsf{P}} \times A^{\mathsf{A}}$ is a finite joint action space; $\Omega = \Omega^{\mathsf{P}} \times \Omega^{\mathsf{A}}$ is a finite joint observation space; $p = (p_h)_{h=0}^{H-1}$ and $\mathbf{r} = (\mathbf{r}_h)_{h=1}^{H}$ are two tuples, each consisting of an element for every time step $h$. Specifically, $p_0 \in \Delta(S \times \Omega)$ is a distribution of the initial state-observation pairs, and each $p_h$, $h \geq 1$, is a transition function $p_h : S \times A \to \Delta(S \times \Omega)$. Each $\mathbf{r}_h = (r_h^{\mathsf{P}}, r_h^{\mathsf{A}})$ is a pair of reward functions $r_h^{\mathsf{P}} : S \times A \to \mathbb{R}$ and $r_h^{\mathsf{A}} : S \times A \to \mathbb{R}$, for the principal and the agent, respectively. W.l.o.g., we assume that all rewards are in $[0, 1]$, and all rewards in the last time step $H$ are 0.

The interaction proceeds as follows. At the beginning, an initial state-observation pair $(s_1, \boldsymbol{\omega}_1) \sim p_0$ is drawn. Then, each time step $h = 1, \dots, H$ involves the following stages.

1. **Observation:** The principal and the agent observe, privately, $\omega_h^{\mathsf{P}}$ and $\omega_h^{\mathsf{A}}$, respectively (but not $s_h$).

2. **Communication:** The principal elicits the agent's observation. The agent reports $\widetilde{\omega}_h^{\mathsf{A}} \in \Omega^{\mathsf{A}}$ (possibly different from $\omega_h^{\mathsf{A}}$). Then, based on $\omega_h^{\mathsf{P}}$ and $\widetilde{\omega}^{\mathsf{A}}$ the principal recommends an action $a_h^{\mathsf{A}}$ for the agent to play. The recommendation is sent to the agent as a coordination signal.

3. **Action:** Based on the information exchange above, the principal and the agent, simultaneously, each perform an action, say $a_h^{\mathsf{P}}$ and $\tilde{a}_h^{\mathsf{A}}$, respectively. (The action $\tilde{a}_h^{\mathsf{A}}$ the agent actually performs may be different from the recommended one $a_h^{\mathsf{A}}$.)

4. **Rewards and next state:** Rewards $r_h^{\mathsf{P}}(s_h, a_h^{\mathsf{P}}, \tilde{a}_h^{\mathsf{A}})$ and $r_h^{\mathsf{A}}(s_h, a_h^{\mathsf{P}}, \tilde{a}_h^{\mathsf{A}})$ are generated for the principal and agent, respectively. The next state is drawn: $s_{h+1} \sim p_h(\cdot \mid s_h, a_h^{\mathsf{P}}, \tilde{a}_h^{\mathsf{A}})$.

Following the general paradigm of principal-agent problems, we consider the principal's *commitment* to a coordination policy. The agent best-responds to the principal's commitment. Both players are *far-sighted* and aim to maximize the sum of their rewards over the $H$ time steps.[1] We take the principal's perspective and the goal, as we will shortly formalize, is to compute the principal's optimal commitment. At a high level, this is a Stackelberg game between the principal and the agent and we aim to compute a Stackelberg equilibrium.

## 2.1 Hindsight Observability

We assume *hindsight observability*, whereby both players observe the full interaction history (including $s_h$, $\boldsymbol{\omega}$, and $\mathbf{a}$) at the *end* of each time step. This condition is essential for circumventing intrinsic computational complexity barriers: without it, our model would directly subsume partially observable MDPs (POMDPs), which is famously known to be PSPACE-hard [36], in which case any efficient algorithms would be hopeless unless P = PSAPCE. (See a further discussion in Appendix C.) Similar assumptions have been adopted in the POMDP literature [24]. Importantly, our model remains highly expressive under hindsight observability, capturing a broad range of relevant subcases, including: scenarios where the state is immediately observable, e.g., repeated games, stochastic games with full state observability [10], as well as scenarios where observations can be interpreted as external parameters generated based on an internal Markovian state observable to both players (e.g., [15, 41]).

Indeed, in many real-world scenarios, it is natural for players to observe each other's actions after they are taken (consider, e.g., repeated rock–paper–scissors). Beyond actions, players may also receive regular updates about each other's private observations. This occurs in particular when each player observes a different part of the state and the full state is revealed at the end of the time step—at which point all private observations effectively become public. For example, traders may each observe local or regional market information that is later aggregated and released publicly; in energy markets, regional grid operators obtain first-hand information about supply and demand within their own regions and later receive system-wide reports covering the entire market; and in an R&D consortium, each firm privately evaluates prototypes or test results and acts on that information during the reporting cycle, after which all results are disclosed to members under agreed rules.

## 2.2 History-dependent Policy

We consider history-dependent policies, which are more general than stationary policies and hence typically yield higher payoffs. For example, to punish the agent for performing a certain action requires a history-dependent policy that remember the agent's action in the previous time step. History-dependent policies are also a natural choice for finite-horizon models, where the memory required to track the history is bounded by the horizon length.

A *history* up to time step $h$ is a sequence $\sigma = \left(s_\ell, \boldsymbol{\omega}_\ell, \widetilde{\omega}_\ell^{\mathsf{A}}, \mathbf{a}_\ell, \tilde{a}_\ell^{\mathsf{A}}\right)_{\ell=1}^{h}$, containing elements in the four stages of each step described above (and we write $\boldsymbol{\omega}_\ell = (\omega_\ell^{\mathsf{P}}, \omega_\ell^{\mathsf{A}})$ and $\mathbf{a}_\ell = (a_\ell^{\mathsf{P}}, a_\ell^{\mathsf{A}})$). We let $\Sigma_h$ denote the set of all sequences till time step $h$, and let $\Sigma = \bigcup_{h=0}^{H} \Sigma_h$, where $\Sigma_0 = \{\varnothing\}$ contain only the empty sequence. Moreover, we denote by $\bar{\Sigma} := S \times \Omega \times \Omega^{\mathsf{A}} \times A \times A^{\mathsf{A}}$ the set of all possible interactions within one time step. We can now write the transition function as $p_h(\cdot \mid \sigma) = p_h(\cdot \mid s_h, \mathbf{a}_h)$ for any given sequence $\sigma \in \Sigma_h$ (specially, $p_0(\cdot \mid \varnothing) = p_0(\cdot)$).

**Principal's Policy**  A history-dependent policy takes the form $\pi : \Sigma \times \Omega \to \Delta(A)$, whereby upon seeing $\sigma$ in the previous steps, observing $\omega^{\mathsf{P}}$, and receiving the agent's report $\widetilde{\omega}^{\mathsf{A}}$ in the current step, the principal draws a joint action $\mathbf{a} = (a^{\mathsf{P}}, a^{\mathsf{A}}) \sim \pi(\sigma; \omega^{\mathsf{P}}, \widetilde{\omega}^{\mathsf{A}})$, sends $a^{\mathsf{A}}$ to the agent as an action recommendation, and performs $a^{\mathsf{P}}$ herself. We denote by $\pi(\mathbf{a} \mid \sigma; \omega^{\mathsf{P}}, \widetilde{\omega}^{\mathsf{A}})$ the probability of each $\mathbf{a}$ in $\pi(\sigma; \omega^{\mathsf{P}}, \widetilde{\omega}^{\mathsf{A}})$.

---

[1]While we assume no discounting, all our results easily extend to discounted rewards.

**Agent's Response** The principal's commitment results in a meta-POMDP for the agent. The agent reacts by playing optimally in this meta-POMDP. When the principal's policy is IC, this simply means responding truthfully. Formally, the agent's strategy can be described by a *deviation plan* $\rho : (\sigma, \omega^A) \mapsto (\widetilde{\omega}^A, f : A^A \to A^A)$, such that given any history $\sigma$ and observation $\omega^A$ in the current step, the agent reports $\widetilde{\omega}^A$ and then plays $\tilde{a}^A = f(a^A)$ if subsequently the principal recommends playing $a^A$. For simplicity, we write $\widetilde{\omega}^A = \rho(\sigma; \omega^A)$ and $\tilde{a}^A = \rho(\sigma; \omega^A, a^A)$. We denote by $\perp$ the special deviation plan where no deviation is made, i.e., $\perp (\sigma; \omega^A) \equiv \omega^A$ and $\perp (\sigma; \omega^A, a^A) \equiv a^A$.

The agent's value (i.e., total reward) induced by a policy $\pi$ and a deviation strategy $\rho$ can be defined recursively via the value function as follows. For every $h = 1, \ldots, H-1$ and $\sigma \in \Sigma_{h-1}$:

$$V_h^{A,\pi,\rho}(\sigma) := \mathbb{E}\left[ r_h^A\left(s, a^P, \tilde{a}^A\right) + V_{h+1}^{A,\pi,\rho}\left(\sigma; s, \boldsymbol{\omega}, \widetilde{\omega}^A, \mathbf{a}, \tilde{a}^A\right) \right], \tag{1}$$

where the expectation is taken over $(s, \boldsymbol{\omega}) \sim p_{h-1}(\cdot \,|\, \sigma)$ and $\mathbf{a} \sim \pi(\cdot \,|\, \sigma, \omega^P, \widetilde{\omega}^A)$; moreover, $\widetilde{\omega}^A = \rho(\sigma^A, \omega^A)$ and $\tilde{a}^A = \rho(\sigma^A, \omega^A, a^A)$, and by assumption $V_H^A(\sigma) \equiv 0$ for the last time step. The principal's value is defined the same way by changing the labels.

Our goal is to find a policy $\pi$ that maximizes the principal's value under the agent's best response:

$$\max_{\pi, \rho} \quad V_1^{P,\pi,\rho}(\varnothing) \tag{2}$$

$$\text{subject to} \quad \rho \in \arg\max_{\rho'} V_1^{A,\pi,\rho'}(\varnothing) \tag{2-1}$$

In other words, we look for $\pi$ and $\rho$ that form a Stackelberg equilibrium. We say that $\pi$ is $\epsilon$-*optimal* if $V_1^{P,\pi,\rho}(\varnothing) \geq V^* - \epsilon$ for some $\rho$ satisfying (2-1), where $V^*$ denotes the optimal value of (2).

As we will demonstrate, under hindsight observability, it is without loss of optimality to consider policies that are IC (incentive compatible), which incentivize $\perp$ as an optimal response of the agent.

**Definition 1** (IC policy). A policy $\pi$ is IC if $V_1^{A,\pi,\perp}(\varnothing) \geq V_1^{A,\pi,\rho}(\varnothing)$ for every possible deviation plan $\rho$ of the agent.

## 3 Computing an Optimal Policy

We compute a near-optimal policy by constructing the *inducible value sets*. The approach is similar to value-based approaches to solving MDPs, which operate by reasoning about the values of the states. However, it involves the following crucial differences:

1. Since we are in a two-player setting, both player's values need to be incorporated. We use a two-dimensional value vector instead of a scalar value.

2. Since we consider history-dependent policies, the game may proceed differently from the same state based on different histories. In this case, a single value (vector) is no longer sufficient for characterizing a state, unlike the case with stationary policies. We use the set consisting of all values can be induced by some valid policy to characterize a state. We refer to these sets are *inducible value sets*.

Moreover, since the actual state of the underlying MDP is not observable at the beginning of each time step, we will instead characterize the inducible value sets of *state-action pairs* $o \in O := S \times A$. Under hindsight observability, these pairs function as the states in a Markovian process, which the players observe before they each decide an action to play.

**Definition 2** (Inducible value set). The inducible value set $\mathcal{V}_h(o) \subseteq \mathbb{R}^2$ of a state-action pair $o \in O := S \times A$ at time step $h$ consists of all vectors $\mathbf{v} = (v^P, v^A)$ such that $v^P = V_h^{P,\pi,\rho}(\sigma)$ and $v^A = V_h^{A,\pi,\rho}(\sigma)$ for some policy $\pi$, deviation plan $\rho \in \arg\max_{\rho'} V_h^{A,\pi,\rho'}(\sigma)$, and sequence $\sigma \in \Sigma_{h-1}$ ending with $o$.

It is straightforward that once we obtain $\mathcal{V}_1(\varnothing)$, then the principal's optimal value in (2) can be computed by solving $\max_{(v^P, v^A) \in \mathcal{V}_1(\varnothing)} v^P$. We present a dynamic programming approach to constructing the inducible sets next. We will later also show that, once we obtain all the inducible sets, every inducible value vector can be efficiently unrolled into a history-dependent policy that induces it, and the policy can be executed efficiently on the fly. Hence, each value vector serves as a compact representation of a fully specified history-dependent policy.

### 3.1 Computing Inducible Value Sets

Instead of computing the exact value sets, we use convex polytopes to approximate them, so as to avoid possible exponential growth of the representational complexity. Let $\widehat{\mathcal{V}}_h(o)$ denote the approximation of $\mathcal{V}_h(o)$ we aim to obtain. Recall that in the last time step all rewards are 0, so trivially we use $\widehat{\mathcal{V}}_H(o) = \mathcal{V}_H(s) = \{(0,0)\}$ for all $o \in O$ as the base case.

**Dynamic Programming**  Now suppose that we have obtained the polytopes $\widehat{\mathcal{V}}_{h+1}(o')$ for all $o' \in O$. We move to time step $h$ and construct each $\widehat{\mathcal{V}}_h(o)$ based on the $\widehat{\mathcal{V}}_{h+1}(o')$'s. Central to the approach is the following characterization, which describes an IC condition at time step $h$: for every $\mathbf{v} \in \mathbb{R}^2$, it holds that $\mathbf{v} \in \mathcal{V}_h(o)$ if and only if there exist a one-step policy $\bar{\pi} : \Omega \to \Delta(A)$ and a set of onward value vectors $\left\{ \mathbf{v}'(\bar{\sigma}) \in \mathbb{R}^2 : \bar{\sigma} \in \bar{\Sigma} \right\}$ that satisfy the following constraints.

1. A **value function constraint** based on (1), which expresses $\mathbf{v}$ via the immediate rewards and onward value vectors $\mathbf{v}'$ to be induced next, assuming truthful response of the agent:

$$\mathbf{v} = \sum_{s,\boldsymbol{\omega},\mathbf{a}} p_{h-1}(s,\boldsymbol{\omega} \,|\, o) \cdot \bar{\pi}(\mathbf{a} \,|\, \boldsymbol{\omega}) \cdot \Big( \mathbf{r}_h(s,\mathbf{a}) + \mathbf{v}'(s,\boldsymbol{\omega},\omega^{\mathsf{A}},\mathbf{a},a^{\mathsf{A}}) \Big), \tag{3}$$

The onward value vectors represent the subsequent part of the principal's commitment, which is contingent on the interaction $(s, \boldsymbol{\omega}, \widetilde{\omega}^{\mathsf{A}}, \mathbf{a}, \tilde{a}^{\mathsf{A}})$ in step $h$. Under the truthful response of the agent, we have $\widetilde{\omega}^{\mathsf{A}} = \omega^{\mathsf{A}}$ and $\tilde{a}^{\mathsf{A}} = a^{\mathsf{A}}$ in (3).

2. **IC constraints**, which ensure that the agent's truthful behavior assumed in (3) is indeed incentivized, where we denote by $p_{h-1}(s, \omega^{\mathsf{P}} \,|\, o, \omega^{\mathsf{A}}) \propto p_{h-1}(s, \boldsymbol{\omega} \,|\, o)$ the conditional probability defined by $p_{h-1}$. For all $\omega^{\mathsf{A}}, \widetilde{\omega}^{\mathsf{A}} \in \Omega^{\mathsf{A}}$,

$$\sum_{s,\omega^{\mathsf{P}},\mathbf{a}} p_{h-1}(s,\omega^{\mathsf{P}} \,|\, o,\omega^{\mathsf{A}}) \cdot \bar{\pi}(\mathbf{a} \,|\, \boldsymbol{\omega}) \cdot \Big( r_h^{\mathsf{A}}(s,\mathbf{a}) + v'^{\mathsf{A}}(s,\boldsymbol{\omega},\omega^{\mathsf{A}},\mathbf{a},a^{\mathsf{A}}) \Big) \geq$$

$$\sum_{a^{\mathsf{A}}} \max_{\tilde{a}^{\mathsf{A}} \in A^{\mathsf{A}}} \sum_{s,\omega^{\mathsf{P}},a^{\mathsf{P}}} p_{h-1}(s,\omega^{\mathsf{P}} \,|\, o,\omega^{\mathsf{A}}) \cdot \bar{\pi}(\mathbf{a} \,|\, \omega^{\mathsf{P}},\widetilde{\omega}^{\mathsf{A}}) \cdot \Big( r_h^{\mathsf{A}}(s,a^{\mathsf{P}},\tilde{a}^{\mathsf{A}}) + v'^{\mathsf{A}}(s,\boldsymbol{\omega},\widetilde{\omega}^{\mathsf{A}},\mathbf{a},\tilde{a}^{\mathsf{A}}) \Big).$$
$$\tag{4}$$

Namely, the constraint says, upon observing $\omega^{\mathsf{A}}$, the agent's expected payoff under their truthful response is at least as much as what they could have obtained, had they: 1) reported a different observation $\widetilde{\omega}^{\mathsf{A}}$, 2) performed a best action $\tilde{a}^{\mathsf{A}}$ in response to every possible recommendation $a^{\mathsf{A}}$ of the principal, and 3) responded optimally in the subsequent time steps (whereby the onward values are given by $\mathbf{v}'$).

3. **Onward value constraints**, which ensures that the onward values given by $\mathbf{v}'$ are also inducible:

$$\mathbf{v}'(s,\boldsymbol{\omega},\widetilde{\omega}^{\mathsf{A}},\mathbf{a},\tilde{a}^{\mathsf{A}}) \in \mathcal{V}_{h+1}(s,a^{\mathsf{P}},\tilde{a}^{\mathsf{A}}) \qquad \text{for all } (s,\boldsymbol{\omega},\widetilde{\omega}^{\mathsf{A}},\mathbf{a},\tilde{a}^{\mathsf{A}}) \in \bar{\Sigma}. \tag{5}$$

The following lemma indicates the correctness of the above characterization.

**Lemma 3.** $\mathbf{v} \in \mathcal{V}_h(o)$ if and only if (3) to (5) hold for some $\bar{\pi} : \Omega \to \Delta(A)$ and $\mathbf{v}' : \bar{\Sigma} \to \mathbb{R}^2$.

Therefore, to decide whether $\mathbf{v} \in \mathcal{V}_h(o)$ amounts to deciding whether the above constraints are satisfied by some $\bar{\pi}$ and $\mathbf{v}'$ (highlighted in blue in the constraints). Note that since the inductive hypothesis assumes an approximation $\widehat{\mathcal{V}}_{h+1}(o')$ instead of the exact set $\mathcal{V}_{h+1}(o')$, we will in fact impose the following *approximate* onward value constraint, instead of the exact version in (5):

$$\mathbf{v}'(s,\boldsymbol{\omega},\widetilde{\omega}^{\mathsf{A}},\mathbf{a},\tilde{a}^{\mathsf{A}}) \in \widehat{\mathcal{V}}_{h+1}(s,a^{\mathsf{P}},\tilde{a}^{\mathsf{A}}) \qquad \text{for all } (s,\boldsymbol{\omega},\widetilde{\omega}^{\mathsf{A}},\mathbf{a},\tilde{a}^{\mathsf{A}}) \in \bar{\Sigma}. \tag{6}$$

**Linearizing (3) and (4)**  The constraint satisfiability problem defined above is non-linear due to the quadratic terms and the maximization operator in (3) and (4). Nevertheless, it can be linearized as long as every polytope $\widehat{\mathcal{V}}_{h+1}(o')$, $o' \in O$, is given by the *half-space representation*, i.e., by linear constraints in the form $\mathbf{H} \cdot \mathbf{x} \leq \mathbf{b}$ for some matrix $\mathbf{H}$ and vector $\mathbf{b}$. Due to the space limit, we leave the details in Appendix B.

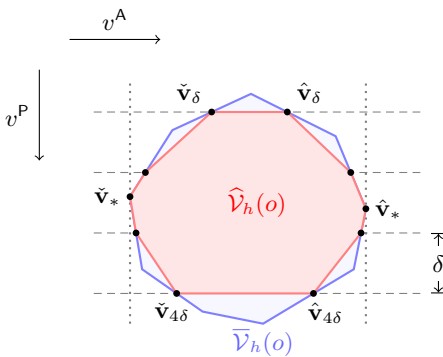

Figure 1: Constructing $\widehat{\mathcal{V}}_h(o)$ as a $\delta$-approximation of $\overline{\mathcal{V}}_h(o)$. The black points constitute $\mathcal{W}$.

---

For $h = H - 1, \ldots, 1$, do the following for all $o \in O$:

1. Plug in (6) the half-space representation of $\widehat{\mathcal{V}}_{h+1}(o')$, $o' \in O$. Then linearize (3) and (4).

2. Discretize the space $[0, H]^2$ into a finite point set (see Lemma 4 for more detail). Check the inducibility of each point $\mathbf{v}$ in this set by solving the linear constraint satisfiability problem defined by (the linearized version of) (3), (4) and (6).

3. Compute $\widehat{\mathcal{V}}_h(o)$ as the convex hull of the inducible points obtained above, in half-space representation.

---

Figure 2: Computing approximate value polytopes via dynamic programming.

**Constructing $\widehat{\mathcal{V}}_h(o)$**  As a result, we obtain a polytope $\mathcal{P}$ defined by a set of linear constraints equivalent to (3), (4) and (6). The projection of $\mathcal{P}$ onto the dimensions of $\mathbf{v}$ is (approximately) $\mathcal{V}_h(o)$. To ensure that the projection can be plugged back into (6) in the next induction step, we need the half-space representation of the projection, too. In particular, we want to eliminate the additional variables in the representation so that only $\mathbf{v}$ remains. (Otherwise, the number of variables may grow exponentially as the induction step increases.) This can be done approximately in polynomial time given that $\mathbf{v}$ is two-dimensional. Roughly speaking, we discretize the box $[0, H]^2$ into a finite set of points (recall that rewards in each time step are bounded in $[0, 1]$, so $[0, H]^2$ contains $\mathcal{V}_h(o)$), check the inducibility of each point, and compute the convex hull of the inducible points in half-space representation. The specific way we discretize the space (see Fig. 1) ensures that IC is satisfied *exactly* (which can otherwise not be achieved by using standard grid-based discretization). The details can be found in the proof of Lemma 4.

Repeating the induction procedure till $h = 1$, we obtain $\widehat{\mathcal{V}}_1(\varnothing)$ as well as a near-optimal value of the principal by solving the LP $\max_{\mathbf{v} \in \widehat{\mathcal{V}}_1(\varnothing)} v^{\mathsf{P}}$. Fig. 2 summarizes this dynamic programming approach.

**Lemma 4.** *For any constant $\epsilon > 0$, it can be computed in time $\mathrm{poly}(|S|\cdot|A|\cdot|\Omega|, H, 1/\epsilon)$ the half-space representations of a set of polytopes $\widehat{\mathcal{V}}_h(o) \subseteq \mathcal{V}_h(o)$, $o \in O \cup \{\varnothing\}$ and $h = 1, \ldots, H$, such that (3), (4) and (6) are satisfiable for every $\mathbf{v} \in \widehat{\mathcal{V}}_h(o)$ and $\max_{\mathbf{v} \in \widehat{\mathcal{V}}_1(\varnothing)} v^{\mathsf{P}} \geq \max_{\mathbf{v} \in \mathcal{V}_1(\varnothing)} v^{\mathsf{P}} - \epsilon$.*

*Proof sketch.* We construct the approximate sets $\widehat{\mathcal{V}}_h(o)$ by backward induction. Assume for all $o$ that $\widehat{\mathcal{V}}_{h+1}(o)$ is defined by polynomially many constraints and is an $\varepsilon$-approximation of $\mathcal{V}_{h+1}(o)$. Let $\overline{\mathcal{V}}_h(o)$ be the feasible set defined by (3), (4) and (6) using $\widehat{\mathcal{V}}_{h+1}(o')$. We discretize the principal's value into *slices* of width $\delta$ (see Fig. 1). We include the following points into a collection $\mathcal{W}$ use the convex hull of $\mathcal{W}$ as $\widehat{\mathcal{V}}_h(o)$: 1) for every line $w$ separating the slices, the two points $\check{\mathbf{v}}_w$ and $\hat{\mathbf{v}}_w$ at the intersection of $w$ and the boundary of $\widehat{\mathcal{V}}_{h+1}(o)$; and 2) the extreme points $\check{\mathbf{v}}_*$ and $\hat{\mathbf{v}}_*$ of $\overline{\mathcal{V}}_h(o)$ minimizing and maximizing the agent's value, respectively. Each of these points can be obtained

Input: a sequence $(\sigma; \omega^{\mathsf{P}}, \widetilde{\omega}^{\mathsf{A}})$, where $\sigma = (s_\ell, \boldsymbol{\omega}_\ell, \widetilde{\omega}_\ell^{\mathsf{A}}, \mathbf{a}_\ell, \tilde{a}_\ell^{\mathsf{A}})_{\ell=1}^{h-1}$.

1. Initialize: $\mathbf{v} \leftarrow \arg\max_{\mathbf{v} \in \widehat{\mathcal{V}}_1(\varnothing)} v^{\mathsf{P}}$ and $o \leftarrow \varnothing$.

2. For $\ell = 1, \ldots, h-1$:

   - Fix $\mathbf{v}$ and $o$, and solve (3), (4) and (6), where we use the polytopes $\widehat{\mathcal{V}}_h(o)$ described in Lemma 4. Let the solution be $\bar{\pi}$ and $\mathbf{v}'$.
   - Update: $\mathbf{v} \leftarrow \mathbf{v}'(s_\ell, \boldsymbol{\omega}_\ell, \widetilde{\omega}_\ell^{\mathsf{A}}, \mathbf{a}_\ell, \tilde{a}_\ell^{\mathsf{A}})$ and $o \leftarrow (s_\ell, a_\ell^{\mathsf{P}}, a_\ell^{\mathsf{A}})$.

3. Output $\pi(\cdot \mid \sigma; \omega^{\mathsf{P}}, \widetilde{\omega}^{\mathsf{A}}) = \bar{\pi}(\cdot \mid \omega^{\mathsf{P}}, \widetilde{\omega}^{\mathsf{A}})$.

Figure 3: Computing a near-optimal policy based on approximations of the value polytopes.

efficiently by solving a polynomial-size LP. The inclusion of the extreme points ensures in particular that we do not miss the agent's extreme values, which is critical for achieving exact IC.

It can be verified that $\widehat{\mathcal{V}}_h(o) \subseteq \overline{\mathcal{V}}_h(o)$ and that any $\mathbf{v} \in \overline{\mathcal{V}}_h(o)$ can be approximated by some $\mathbf{x} \in \widehat{\mathcal{V}}_h(o)$ with $x^{\mathsf{A}} = v^{\mathsf{A}}$ and $x^{\mathsf{P}} \geq v^{\mathsf{P}} - \delta$. So $\widehat{\mathcal{V}}_h(o)$ is a $\delta$-approximation of $\overline{\mathcal{V}}_h(o)$ and $x^{\mathsf{A}} = v^{\mathsf{A}}$ ensures exact IC for the agent. Since $\overline{\mathcal{V}}_h(o)$ itself is an $\varepsilon$-approximation of $\mathcal{V}_h(o)$, we obtain an $(\varepsilon + \delta)$-approximation overall. Choosing $\delta = \epsilon/H$ and noting the trivial base case $\widehat{\mathcal{V}}_H(o) = \{(0,0)\}$, the construction yields $\widehat{\mathcal{V}}_1(\varnothing)$ as an $\epsilon$-approximation of $\mathcal{V}_1(\varnothing)$, computable in time $\mathrm{poly}(|S| \cdot |A| \cdot |\Omega|, H, 1/\epsilon)$. □

### 3.2 Unrolling the Optimal Policy

The above procedure yields the maximum inducible value of the principal but not yet an optimal policy that achieves this value. We next demonstrate how to compute an optimal policy based on $\widehat{\mathcal{V}}_1(\varnothing)$. Rather than obtaining an explicit description of a history-dependent policy $\pi$—which would be exponentially large as the policy specifies a distribution for each possible sequence—we present an efficient procedure that computes the distribution $\pi(\cdot \mid \sigma; \omega^{\mathsf{P}}, \widetilde{\omega}^{\mathsf{A}})$ for any given sequence $(\sigma; \omega^{\mathsf{P}}, \widetilde{\omega}^{\mathsf{A}})$. This means that, when playing the game, the principal can compute an optimal policy on the fly based on the realized history.

We use a forward computation procedure presented in Fig. 3. Starting from time step 1, the procedure repeatedly computes a one-step policy $\bar{\pi}$ and a set of onward vectors, to induce the target value vector $\mathbf{v}$. The onward vectors define the target values to be induced in the next time step, contingent on the interaction in the current, which is given by $\sigma$. Hence, the target vector is updated to one of the onward vectors according $\sigma$ at the end of each iteration. In other words, in each time step, we expand the target vector into a set of onward vectors, and then select one of them as the next target vector according to the realized interaction given by $\sigma$. This leads to the following main result.

**Theorem 5.** *There exists an $\epsilon$-optimal IC policy $\pi$ such that, for any given sequence $(\sigma; \omega^{\mathsf{P}}, \widetilde{\omega}^{\mathsf{A}}) \in \Sigma \times \Omega$, the distribution $\pi(\cdot \mid \sigma; \omega^{\mathsf{P}}, \widetilde{\omega}^{\mathsf{A}})$ can be computed in time $\mathrm{poly}(|S| \cdot |A| \cdot |\Omega|, H, 1/\epsilon)$.*

## 4 Learning to Commit

We now turn to an episodic online learning setting where the transition model $p: S \times A \to \Delta(S \times \Omega)$ is not known to the players beforehand. Let there be $T$ episodes. At the beginning of each episode, the principal commits to a new policy based on the outcomes of the previous episodes. Each episode proceeds in $H$ time steps the same way as the model defined in Section 2.

We present a learning algorithm that guarantees sublinear regrets for both players under hindsight observability. The algorithm is *centralized* and relies on the agent behaving truthfully. It does not guarantee exact IC during the course of learning but IC in the limit when the number of episodes approaches infinity. Indeed, since the model is unknown to both players, IC in the limit is a more relevant concept as the agent cannot decide how to optimally deviate from their truthful response, either. In this case, the sublinear regret the algorithm guarantees for the agent should in many

scenarios be sufficient for incentivizing for the agent to participate and follow the centralized learning protocol.

The players' regrets are defined as follows:

$$\text{Reg}^{\mathsf{P}} = \sum_{t=1}^{T} \left( V^* - V_1^{\mathsf{P},\pi_t,\perp}(\varnothing) \right) \quad \text{and} \quad \text{Reg}^{\mathsf{A}} = \sum_{t=1}^{T} \left( \max_{\rho} V_1^{\mathsf{A},\pi_t,\rho}(\varnothing) - V_1^{\mathsf{A},\pi_t,\perp}(\varnothing) \right),$$

where $V^*$ is the optimal value of (2) and $\pi_t$ denotes the policy the principal commits to in the $t$-th episode. In words, the principal's regret $\text{Reg}^{\mathsf{P}}$ is defined with respect to the optimal policy under the true model. The agent's regret $\text{Reg}^{\mathsf{A}}$ is defined with respect to his optimal response to each $\pi_t$, which is a dynamic regret as the benchmark changes across the episodes.

### 4.1 Learning Algorithm

**Reward-free Exploration**  Our learning algorithm is based on *reward-free exploration*, which is an RL paradigm where learning happens before a reward function is provided [21]. It has been shown in a series of works that efficient learning is possible under this paradigm [21, 23, 30]. In particular, we will use the sample complexity bound in Lemma 6. At a high level, our algorithm proceeds by first conducting reward-free exploration to learn a sufficiently accurate estimate of the true model. Based on the estimate we then solve a relaxed version of the policy optimization problem (2) to obtain a policy. Using this policy in the remaining episodes guarantees sublinear regret for both players.

**Lemma 6** ([21, Lemma 3.6 restated])**.** *Consider an (single-player) MDP $(S, A, p)$ (without any reward function specified) with horizon length $H$. There exists an algorithm which learns a model $\widehat{p}$ after $\widetilde{\mathcal{O}}\left( \frac{H^5 |S|^2 \cdot |A|}{\delta^2} \right)$ episodes of exploration, such that with probability at least $1 - q$, for any reward function $r$ and policy $\pi$, it holds that $\left| V_1^{r,\pi}(s) - \widehat{V}_1^{r,\pi}(s) \right| \leq \delta/2$ for all states $s$, where $V_1^{r,\pi}$ and $\widehat{V}_1^{r,\pi}$ denote the value functions under reward function $r$ and models $p$ and $\widehat{p}$, respectively.*[2]

With the above result, we can learn a model $\widehat{p}$ for our purpose. In what follows, we let $\widehat{V}_h^{\mathsf{P},\pi,\rho}$ and $\widehat{V}_h^{\mathsf{A},\pi,\rho}$ denote the players' value functions in model $\widehat{p}$ (i.e., by replacing $p$ in (1) with $\widehat{p}$). Lemma 7 then translates Lemma 6 to our setting. Note that under hindsight observability the process facing the principal and the agent jointly during the learning process is effectively an MDP, where the effective state space is $O \times \Omega$. An effective state, say $\theta = (s, \mathbf{a}, \boldsymbol{\omega})$, consists of the state-action pair $(s, \mathbf{a})$ in the previous step and the observations $\boldsymbol{\omega}$ in the current. When a joint action $\mathbf{a}'$ is performed, $\theta$ transitions to $\theta' = (s', \mathbf{a}', \boldsymbol{\omega}')$ with probability $p_{h-1}(s', \boldsymbol{\omega}' \mid s, \mathbf{a})$.

**Lemma 7.** *A model $\widehat{p}$ can be learned after $\widetilde{\mathcal{O}}\left( H^5 |S|^2 |A|^3 |\Omega|^2 / \delta^2 \right)$ episodes of exploration, such that $\left| V_1^{\mathsf{A},\pi,\rho}(\varnothing) - \widehat{V}_1^{\mathsf{A},\pi,\rho}(\varnothing) \right| \leq \delta/2$ and $\left| V_1^{\mathsf{P},\pi,\rho}(\varnothing) - \widehat{V}_1^{\mathsf{P},\pi,\rho}(\varnothing) \right| \leq \delta/2$ with probability at least $1 - q$ for any policy $\pi$ and deviation plan $\rho$.*

Therefore, the value functions change smoothly as the learned model $\widehat{p}$ approaches $p$. However, this smoothness is insufficient for deriving a sublinear bound on the principal's regret because of the agent's incentive constraints in our problem. Roughly speaking, the set of IC policies does not change smoothly with $\widehat{p}$, even though the value functions do. Hence, even an infinitesimal difference between $\widehat{p}$ and $p$ may lead to a jump between the IC policy sets under these two models and, in turn, a gap between the values of the optimal policies.

**Approximate IC Relaxation**  To deal with this issue, we relax the incentive constraints, allowing small violations to the constraints. Such violations are inevitable if we aim to achieve a near-optimal value under the true model $p$ but only know an estimate $\widehat{p}$ of the true model. On the positive side, given the sublinear regret guarantee for the agent, the violation diminishes with the number of episodes. We define $\delta$-IC policies below.

---

[2]The notation $\widetilde{\mathcal{O}}$ omits logarithmic factors. In the original statement of Jin et al. [21], $\pi$ is non-stationary (time-dependent) but independent of the history. However, the proof of the lemma also applies to history-dependent policies. The dependence on $H$ in the sample complexity can be further improved with better reward-free exploration algorithms [23, 30], but this is not a focus of ours.

**Definition 8** ($\delta$-IC policy). A policy $\pi$ is $\delta$-IC (w.r.t. model $\widehat{p}$) if $\widehat{V}_1^{\mathsf{A},\pi,\perp}(\varnothing) \geq \widehat{V}_1^{\mathsf{A},\pi,\rho}(\varnothing) - \delta$ for every possible deviation plan $\rho$ of the agent. A $\delta$-IC policy is said to be $\epsilon$-optimal if $\widehat{V}_1^{\mathsf{P},\pi,\perp}(\varnothing) \geq V^* - \epsilon$, where $V^*$ is the optimal value of (2) (under $p$).

That is, in response to a $\delta$-IC policy, the agent can improve his overall expected payoff by no more than $\delta$ if he deviates from the truthful response. We assume that the agent will not deviate for such a small benefit, and we evaluate the value of a $\delta$-IC policy based on the agent's truthful response. This is how the $\epsilon$-optimality is defined above, where we compare against the optimal value $V^*$ in (2), which is obtained under a more stringent setting without any relaxation of the agent's incentive. In other words, we relax the feasible space and compare the solution obtained in this relaxed space with the optimum over the smaller original feasible space. Such relaxations are common in the optimization literature, and they are crucial for resolving the non-smooth issue.

Let $\widehat{\Pi}_\delta$ and $\Pi_\delta$ denote the set of $\delta$-IC policies under $\widehat{p}$ and $p$, respectively. The relaxation immediately results in $\widehat{\Pi}_\delta \supseteq \Pi_0$ for the model $\widehat{p}$ stated in Lemma 7. As a result, optimizing over $\widehat{\Pi}_\delta$ ensures that the optimal value yielded is as much (up to a small error) as the optimal value $V^*$ over $\Pi_0$. Meanwhile, the value loss introduced by this relaxation for the agent is also small (bounded by $\delta$).

With the above results, our learning algorithm proceeds as follows.

---

1. Run reward-free exploration to obtain a model $\widehat{p}$ as stated in Lemma 7.
2. Compute a $\delta$-optimal $\delta$-IC policy in $\widehat{p}$ and use it in the remaining episodes.

---

The near-optimal policy in Step 2 can be computed efficiently according to Lemma 9, via an approach similar to the one in Section 3.1. This gives an efficient algorithm with sublinear regrets for both players. We present Theorem 10.

**Lemma 9.** *There exists an $\epsilon$-optimal $\delta$-IC policy $\pi$ such that, for any given sequence $(\sigma; \omega^{\mathsf{P}}, \widetilde{\omega}^{\mathsf{A}}) \in \Sigma \times \Omega$, the distribution $\pi(\cdot \mid \sigma; \omega^{\mathsf{P}}, \widetilde{\omega}^{\mathsf{A}})$ can be computed in time $\mathrm{poly}(|S|\cdot|A|\cdot|\Omega|, H, 1/\epsilon, \log(1/\delta))$.*

**Theorem 10.** *There exists an algorithm that guarantees regret $\widetilde{\mathcal{O}}(\zeta^{1/3} T^{2/3})$ for both players with probability $1 - q$, where $\zeta = H^5 |S|^2 |A|^3 |\Omega|^2$. The computation involved in implementing the algorithm takes time $\mathrm{poly}(|S|\cdot|A|\cdot|\Omega|, H, T)$.*

## 5   Conclusion

We studied a stochastic principal-agent framework and presented efficient computation and learning algorithms. Our model can be further extended to the setting with $n$ agents. The algorithms we presented remain efficient for any constant $n$ if approximate IC solutions are considered. Computing optimal exact IC policies for $n$ agents remain an interesting open question, as our discretization method, which operates by slicing the space, does not generalize to $n$ agents. When $n$ is not a constant, representing games in normal-form requires space exponential in $n$, so more succinct representations are typically considered. However, it is known that in succinctly represented games even to compute an optimal correlated equilibrium in one-shot games may be NP-hard [35].) Our results indicate how a policy designer might interact with agents optimally. In particular implementations, the designer's incentives may not be aligned with societal benefits. In these cases, a careful analysis of the incentives and their moral legitimacy must be considered. Besides this, since the paper is theory focused, we do not feel any other potential impacts must be specifically highlighted here.

## Acknowledgement

The work of Goran Radanovic was funded by the Deutsche Forschungsgemeinschaft (DFG, German Research Foundation) – project number 467367360.

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

# A Omitted Proofs

## A.1 Omitted Proofs in Section 3

For simplicity, we write $\vec{V}_h^{\pi,\rho}(\sigma) = \left( V_h^{\mathsf{P},\pi,\rho}(\sigma), V_h^{\mathsf{A},\pi,\rho}(\sigma) \right)$ in the following proofs.

**Lemma 3.** $\mathbf{v} \in \mathcal{V}_h(o)$ *if and only if* (3) *to* (5) *hold for some* $\bar{\pi} : \Omega \to \Delta(A)$ *and* $\mathbf{v}' : \bar{\Sigma} \to \mathbb{R}^2$.

*Proof.* First, consider the "only if" direction of the statement. Suppose that $\mathbf{v} \in \mathcal{V}_h(o)$. By definition, we have $\mathbf{v} = \vec{V}_h^{\pi,\rho}(o)$ for some $\pi$, $\rho \in \arg\max_{\rho'} \vec{V}_h^{\mathsf{A},\pi,\rho'}(\sigma)$, and $\sigma \in \Sigma_{h-1}$ ending with $o$. According to a standard revelation principle argument, we can assume w.l.o.g. that $\rho$ is IC in step $h$. Hence, by (1), we have

$$\mathbf{v} = \sum_{s,\boldsymbol{\omega},\mathbf{a}} p_{h-1}(s,\boldsymbol{\omega} \,|\, o) \cdot \pi(\mathbf{a} \,|\, \sigma; \boldsymbol{\omega}) \cdot \left( \mathbf{r}_h(s,\mathbf{a}) + \vec{V}_{h+1}^{\pi,\rho}(\sigma; s, \boldsymbol{\omega}, \omega^{\mathsf{A}}, \mathbf{a}, a^{\mathsf{A}}) \right). \tag{7}$$

Letting $\bar{\pi}(\mathbf{a} \,|\, \boldsymbol{\omega}) = \pi(\mathbf{a} \,|\, \sigma; \boldsymbol{\omega})$ for every $\boldsymbol{\omega} \in \Omega$, and $\mathbf{v}'(\bar{\sigma}) = \vec{V}_{h+1}^{\pi,\rho}(\sigma; \bar{\sigma})$ for every $\bar{\sigma} \in \bar{\Sigma}$, we establish (3). (Recall that $\bar{\Sigma} := S \times \Omega \times \Omega^{\mathsf{A}} \times A \times A^{\mathsf{A}}$ denotes the set of all possible interactions within one time step.)

Since $\rho$ is IC in step $h$, (4) also follows immediately: the agent cannot benefit from any possible deviation. By definition, we have $\mathbf{v}'(\bar{\sigma}) = \vec{V}_{h+1}^{\pi,\rho}(\sigma; \bar{\sigma}) \in \mathcal{V}_{h+1}(o')$ for every $\bar{\sigma} \in \bar{\Sigma}$ that contains $o'$, so (5) holds.

Now consider the "if" direction. Suppose that (3) to (5) hold for some $\bar{\pi}$ and $\mathbf{v}'$. Pick arbitrary $\sigma \in \Sigma_{h-1}$ that ends with $o$. Consider a policy $\pi$ such that: $\pi(\mathbf{a} \,|\, \sigma; \boldsymbol{\omega}) = \bar{\pi}(\mathbf{a} \,|\, \boldsymbol{\omega})$ for all $\boldsymbol{\omega} \in \Omega$, and $\pi(\mathbf{a} \,|\, \sigma; \bar{\sigma}; \boldsymbol{\omega}) = \pi'(\mathbf{a} \,|\, \sigma; \bar{\sigma}; \boldsymbol{\omega})$ for all $\bar{\sigma} \in \bar{\Sigma}$ and $\boldsymbol{\omega} \in \Omega$, where $\pi'$ is an arbitrary policy that induces $\mathbf{v}'(\bar{\sigma})$ for every $\bar{\sigma}$ (which exists given (5)). Namely, $\pi$ is the same as $\bar{\pi}$ in step $h$ and switches to $\pi'$ in the subsequent steps. Given (4), the agent cannot benefit from any deviation at step $h$, so (3) gives the players' values for $\pi$ and an optimal deviation plan of the agent. Hence, $\mathbf{v} \in \mathcal{V}_h(o)$. $\square$

**Lemma 4.** *For any constant* $\epsilon > 0$, *it can be computed in time* $\mathrm{poly}(|S| \cdot |A| \cdot |\Omega|, H, 1/\epsilon)$ *the half-space representations of a set of polytopes* $\widehat{\mathcal{V}}_h(o) \subseteq \mathcal{V}_h(o)$, $o \in O \cup \{\varnothing\}$ *and* $h = 1, \ldots, H$, *such that* (3), (4) *and* (6) *are satisfiable for every* $\mathbf{v} \in \widehat{\mathcal{V}}_h(o)$ *and* $\max_{\mathbf{v} \in \widehat{\mathcal{V}}_1(\varnothing)} v^{\mathsf{P}} \geq \max_{\mathbf{v} \in \mathcal{V}_1(\varnothing)} v^{\mathsf{P}} - \epsilon$.

*Proof.* Throughout the proof, we say that the polytope $\widehat{\mathcal{V}}_h(o)$ is an $\varepsilon$-*approximation of* $\mathcal{V}_h(o)$ if and only if:

- $\widehat{\mathcal{V}}_h(o) \subseteq \mathcal{V}_h(o)$, and

- for every $\mathbf{v} \in \mathcal{V}_h(o)$, there exists $\mathbf{v}' \in \widehat{\mathcal{V}}_h(o)$ such that $v'^{\mathsf{P}} \geq v^{\mathsf{P}} - \varepsilon$ and $v'^{\mathsf{A}} = v^{\mathsf{A}}$.

We will show that an $\epsilon$-approximation $\widehat{\mathcal{V}}_1(\varnothing)$ of $\mathcal{V}_1(\varnothing)$ can be computed efficiently, so that $\max_{\mathbf{v} \in \widehat{\mathcal{V}}_1(\varnothing)} v^{\mathsf{P}} \geq \max_{\mathbf{v} \in \mathcal{V}_1(\varnothing)} v^{\mathsf{P}} - \epsilon$ follows readily.[3] Meanwhile, we also show that the polytopes we compute ensures that (3), (4) and (6) are satisfiable for every $\mathbf{v} \in \widehat{\mathcal{V}}_h(o)$.

We now prove by induction. The key is the following induction step. Suppose that the following conditions hold for all $o \in O$:

1. $\widehat{\mathcal{V}}_{h+1}(o)$ is defined by $\mathcal{O}(H/\delta)$ many linear constraints.

2. $\widehat{\mathcal{V}}_{h+1}(o)$ is an $\varepsilon$-approximation of $\mathcal{V}_{h+1}(o)$.

---

[3] In the definition of $\epsilon$-approximation, we require additionally that the projections of $\widehat{\mathcal{V}}_h(o)$ and $\mathcal{V}_h(o)$ onto the dimension of $v^{\mathsf{A}}$ are the same (i.e., $v'^{\mathsf{A}} = v^{\mathsf{A}}$), so that the approximation compromises only on the principal's value. This is crucial for ensuring exact IC and smooth changes of the approximation throughout the induction process we present below.

We show that, given the above conditions, for every $o \in O$ we can compute in time polynomial in $1/\delta$ a polytope $\widehat{\mathcal{V}}_h(o)$ (in half-space representation) that satisfies the above conditions (for $h$), with an approximation factor $\varepsilon' = \varepsilon + \delta$ in the second condition. Once this holds, picking $\delta = \epsilon/H$ then gives, by induction, that $\widehat{\mathcal{V}}_1(\varnothing)$ is an $\epsilon$-approximation of $\mathcal{V}_1(\varnothing)$ (where $\epsilon$ is the target constant in the statement of the lemma). Note that as a based case, $\{(0,0)\}$ is readily a $0$-approximation of $\mathcal{V}_H(o)$ and can be defined by three linear constraints.

We proceed as follows. For every $o \in O$, let $\overline{\mathcal{V}}_h(o)$ denote the set of vectors $\mathbf{v}$ satisfying (3), (4) and (6).[4] We follow the algorithm presented in Fig. 2 and discretize $[0, H]^2$ to construct $\widehat{\mathcal{V}}_h(o)$. Specifically, we slice the space along the dimension of the principal's value. We compute the intersection points of the slice lines and (the boundary of) $\overline{\mathcal{V}}_h(o)$, and construct $\widehat{\mathcal{V}}_h(o)$ as the convex hull of the intersection points to approximate $\overline{\mathcal{V}}_h(o)$. Specifically, let $W = \{0, \delta, 2\delta, \ldots, H - \delta, H\}$ contain the principal's values on the slice lines we use, and let $\mathcal{W}$ be the set consisting of the following points.

- First, for each $w \in W$, the two intersection points of the slice line at $w$ and $\overline{\mathcal{V}}_h(o)$:

$$\check{\mathbf{v}}_w \in \arg\min_{\mathbf{v} \in \overline{\mathcal{V}}_h(o): v^{\mathsf{P}} = w} v^{\mathsf{A}} \quad \text{and} \quad \hat{\mathbf{v}}_w \in \arg\max_{\mathbf{v} \in \overline{\mathcal{V}}_h(o): v^{\mathsf{P}} = w} v^{\mathsf{A}}.$$

- Moreover, two vertices of $\overline{\mathcal{V}}_h(o)$ with the minimum and maximum values for the agent:

$$\check{\mathbf{v}}_* \in \arg\min_{\mathbf{v} \in \overline{\mathcal{V}}_h(o)} v^{\mathsf{A}} \quad \text{and} \quad \hat{\mathbf{v}}_* \in \arg\max_{\mathbf{v} \in \overline{\mathcal{V}}_h(o)} v^{\mathsf{A}}.$$

If there are multiple maximum (or minimum) vertices, we pick an arbitrary one.

An illustration is given in Fig. 1.

It shall be clear that the choice of these points ensures that we can approximate any inducible value vector with at most $\delta$ compromise on the principal's value and no compromise on the agent's. (In particular, the inclusion of $\check{\mathbf{v}}_*$ and $\hat{\mathbf{v}}_*$ ensures that we do not miss the agent's extreme values that may not be attained at any of the slice lines.) All the points can be computed efficiently by solving LPs that minimizes (or maximizes) $v^{\mathsf{A}}$ (where we also treat $\mathbf{v}$ as variables in addition to the other variables), subject to the linearized version of (3), (4) and (6), and additionally $v^{\mathsf{P}} = w$ when we compute $\check{\mathbf{v}}_w$ or $\hat{\mathbf{v}}_w$. The hypothesis that $\widehat{\mathcal{V}}_{h+1}(o)$ is defined by $\mathcal{O}(H/\delta)$ linear constraints ensures that all the LPs are polynomial sized and hence can be solved efficiently.

We then compute $\widehat{\mathcal{V}}_h(o)$ by taking the convex hull of $\mathcal{W}$. Given that the space is two-dimensional, this can be done efficiently via standard algorithms in computational geometry (e.g., Chan's algorithm [9]). This way, the first condition in the inductive hypothesis holds for $\widehat{\mathcal{V}}_h(o)$ because $\widehat{\mathcal{V}}_h(o)$ has at most $\mathcal{O}(H/\delta)$ vertices while it is in $\mathbb{R}^2$. Meanwhile, $\widehat{\mathcal{V}}_h(o)$ is an $\delta$-approximation of $\overline{\mathcal{V}}_h(o)$ according to the following arguments.

**Claim 1.** $\widehat{\mathcal{V}}_h(o)$ is an $\delta$-approximation of $\overline{\mathcal{V}}_h(o)$.

*Proof of Claim 1.* First, since $\mathcal{W} \subseteq \overline{\mathcal{V}}_h(o)$ by construction, $\widehat{\mathcal{V}}_h(o) \subseteq \overline{\mathcal{V}}_h(o) \subseteq \mathcal{V}_h(o)$ holds readily. It remains to show that for any $\mathbf{v} \in \overline{\mathcal{V}}_h(o)$ there exists $\mathbf{x} \in \widehat{\mathcal{V}}_h(o)$ such that $x^{\mathsf{A}} = v^{\mathsf{A}}$ and $x^{\mathsf{P}} \geq v^{\mathsf{P}} - \delta$.

Let $\mathcal{B} = \{\mathbf{v}' \in \mathbb{R}^2 : i\delta \leq v'^{\mathsf{P}} \leq (i+1)\delta\}$ be the band between two slice lines that contains $\mathbf{v}$. Consider the relation between $v^{\mathsf{A}}$ and the agent's minimum and maximum values attained at $\mathcal{W} \cap \mathcal{B}$. There can be the following possibilities.

- Case 1. $v^{\mathsf{A}}$ lies in between the minimum and maximum values, i.e.,

$$\min_{\mathbf{v}' \in \mathcal{W} \cap \mathcal{B}} v'^{\mathsf{A}} \leq v^{\mathsf{A}} \leq \max_{\mathbf{v}' \in \mathcal{W} \cap \mathcal{B}} v'^{\mathsf{A}}.$$

  This means that there must be a point $\mathbf{x} \in \mathrm{ConvexHull}(\mathcal{W} \cap \mathcal{B})$ such that $x^{\mathsf{A}} = v^{\mathsf{A}}$. We have $\mathbf{x} \in \mathrm{ConvexHull}(\mathcal{W} \cap \mathcal{B}) \subseteq \mathcal{B}$. So both $\mathbf{v}$ and $\mathbf{x}$ are inside $\mathcal{B}$. According to the definition of $\mathcal{B}$, this means $x^{\mathsf{P}} \geq v^{\mathsf{P}} - \delta$, as desired.

---

[4]Note that $\overline{\mathcal{V}}_h(o)$ is different from $\mathcal{V}_h(o)$: the latter, according to Lemma 3, is defined by (3) to (5), where (5) uses the exact value sets $\mathcal{V}_{h+1}(o')$, unlike the approximate ones $\widehat{\mathcal{V}}_{h+1}(o')$ in (6).

- Case 2. $v^{\mathsf{A}} < \min_{\mathbf{v}' \in \mathcal{W} \cap \mathcal{B}} v'^{\mathsf{A}}$. In this case, it must be that $\check{\mathbf{v}}_* \notin \mathcal{B}$ (otherwise, $\min_{\mathbf{v}' \in \mathcal{W} \cap \mathcal{B}} v'^{\mathsf{A}} = \check{v}_*^{\mathsf{A}} \le v^{\mathsf{A}}$). Now that $\mathbf{v} \in \mathcal{B}$, the line segment between $\mathbf{v}$ and $\check{\mathbf{v}}_*$ must intersect with the boundary of $\mathcal{B}$ (i.e., one of the slice lines) at some point $\mathbf{y}$. We have $y^{\mathsf{A}} \le v^{\mathsf{A}}$ (because $\check{v}_*^{\mathsf{A}} \le v^{\mathsf{A}}$ by definition) and $\mathbf{y} \in \overline{\mathcal{V}}_h(o)$ (because $\mathbf{v}, \check{\mathbf{v}}_* \in \overline{\mathcal{V}}_h(o)$). Pick $\check{\mathbf{v}}_w$ where $w = y^{\mathsf{P}}$. By definition $\check{v}_w^{\mathsf{A}} \le y^{\mathsf{A}}$. It follows that

$$\check{v}_w^{\mathsf{A}} \le y^{\mathsf{A}} \le v^{\mathsf{A}} < \min_{\mathbf{v}' \in \mathcal{W} \cap \mathcal{B}} v'^{\mathsf{A}}.$$

  This is a contradiction because we have $\check{\mathbf{v}}_w \in \mathcal{W} \cap \mathcal{B}$ as $\mathbf{y}$ is on the boundary of $\mathcal{B}$.

- Case 3. $v^{\mathsf{A}} > \max_{\mathbf{v}' \in \mathcal{W} \cap \mathcal{B}} v'^{\mathsf{A}}$. An argument similar to that for Case 2 implies that this case is not possible, either.

Hence, only Case 1 is possible, where a desired point $\mathbf{x}$ exists. The claim then follows. $\square$

The fact that $\widehat{\mathcal{V}}_h(o) \subseteq \overline{\mathcal{V}}_h(o)$ also implies that (3), (4) and (6) are satisfiable for every $\mathbf{v} \in \widehat{\mathcal{V}}_h(o)$, as they are for every $\mathbf{v} \in \overline{\mathcal{V}}_h(o)$. We next confirm that $\widehat{\mathcal{V}}_h(o)$ is eventually an $(\varepsilon + \delta)$-approximation of $\mathcal{V}_h(o)$. Indeed, now Claim 1 indicates that $\widehat{\mathcal{V}}_h(o)$ is an $\delta$-approximation of $\overline{\mathcal{V}}_h(o)$, so $\widehat{\mathcal{V}}_h(o)$ is an $(\varepsilon + \delta)$-approximation of $\mathcal{V}_h(o)$ as long as $\overline{\mathcal{V}}_h(o)$ is an $\varepsilon$-approximation of $\mathcal{V}_h(o)$.

To see that $\overline{\mathcal{V}}_h(o)$ is an $\varepsilon$-approximation, consider an arbitrary $\mathbf{v} \in \mathcal{V}_h(o)$. By Lemma 3, $\mathbf{v}$ can be induced by some $\bar{\pi}$ and $\mathbf{v}'$ satisfying (3) to (5). By assumption, every $\widehat{\mathcal{V}}_{h+1}(o')$ is an $\varepsilon$-approximation of $\mathcal{V}_{h+1}(o')$, so for every onward vector $\mathbf{v}'(\bar{\sigma}) \in \mathcal{V}_{h+1}(o')$, there exists a vector $\tilde{\mathbf{v}}'(\bar{\sigma}) \in \widehat{\mathcal{V}}_{h+1}(o')$ such that $\tilde{v}'^{\mathsf{P}}(\bar{\sigma}) \ge v'^{\mathsf{P}}(\bar{\sigma}) - \varepsilon$ and $\tilde{v}'^{\mathsf{A}}(\bar{\sigma}) = v'^{\mathsf{A}}(\bar{\sigma})$. Using $\tilde{\mathbf{v}}'$ instead of $\mathbf{v}'$, the same policy $\bar{\pi}$ then induces a vector $\tilde{\mathbf{v}} \in \widehat{\mathcal{V}}_h(o)$ to approximate $\mathbf{v}$. Indeed, the agent's values are exactly the same under $\tilde{\mathbf{v}}'$ and $\mathbf{v}'$, so the same response of the agent can be incentivized. This is why we require the approximation to not compromise on the agent's value. Moreover, according to (3), the overall difference between $\tilde{v}^{\mathsf{P}}$ and $v^{\mathsf{P}}$ is at most $\varepsilon$ because it holds for the coefficients that $\sum_{s,\boldsymbol{\omega},\mathbf{a}} p_{h-1}(s, \boldsymbol{\omega} \mid o) \cdot \bar{\pi}(\mathbf{a} \mid \boldsymbol{\omega}) = 1$. As a result, $\tilde{v}^{\mathsf{P}} \ge v^{\mathsf{P}} - \varepsilon$ and $\overline{\mathcal{V}}_h(o)$ is an $\varepsilon$-approximation of $\mathcal{V}_h(o)$.

Hence, the inductive hypothesis holds for $h$. By induction, $\widehat{\mathcal{V}}_1(\varnothing)$ is an $\delta H$-approximation of $\mathcal{V}_1(\varnothing)$. Since $\delta H = \epsilon$, we get that $\max_{\mathbf{v} \in \widehat{\mathcal{V}}_1(\varnothing)} v^{\mathsf{P}} \ge \max_{\mathbf{v} \in \mathcal{V}_1(\varnothing)} v^{\mathsf{P}} - \epsilon$. $\square$

**Theorem 5.** *There exists an $\epsilon$-optimal IC policy $\pi$ such that, for any given sequence $(\sigma; \omega^{\mathsf{P}}, \widetilde{\omega}^{\mathsf{A}}) \in \Sigma \times \Omega$, the distribution $\pi(\cdot \mid \sigma; \omega^{\mathsf{P}}, \widetilde{\omega}^{\mathsf{A}})$ can be computed in time $\mathrm{poly}(|S| \cdot |A| \cdot |\Omega|, H, 1/\epsilon)$.*

*Proof.* Consider the algorithm presented in Fig. 3. The outputs of the algorithm over all possible input sequences $(\sigma; \omega^{\mathsf{P}}, \widetilde{\omega}^{\mathsf{A}}) \in \Sigma \times \Omega$ specify a policy $\pi$. The polynomial running time of the algorithm for computing each $\pi(\cdot \mid \sigma; \omega^{\mathsf{P}}, \widetilde{\omega}^{\mathsf{A}})$ follows by noting that it runs by solving at most $H$ linear constraint satisfiability problems.

It remains to argue that $\pi$ is IC and $\epsilon$-optimal. Indeed, by Lemma 4 and an inductive argument, $\pi$ is IC at each time step $h$ and induces the corresponding values encoded in $\mathbf{v}'$ as the expected onward values. The $\epsilon$-optimality of $\pi$ follows given the condition $\max_{\mathbf{v} \in \widehat{\mathcal{V}}_1(\varnothing)} v^{\mathsf{P}} \ge \max_{\mathbf{v} \in \mathcal{V}_1(\varnothing)} v^{\mathsf{P}} - \epsilon$ stated in Lemma 4 (and the choice of the initial $\mathbf{v}$ in Fig. 3). $\square$

### A.2 Omitted Proofs in Section 4

**Lemma 9.** *There exists an $\epsilon$-optimal $\delta$-IC policy $\pi$ such that, for any given sequence $(\sigma; \omega^{\mathsf{P}}, \widetilde{\omega}^{\mathsf{A}}) \in \Sigma \times \Omega$, the distribution $\pi(\cdot \mid \sigma; \omega^{\mathsf{P}}, \widetilde{\omega}^{\mathsf{A}})$ can be computed in time $\mathrm{poly}(|S| \cdot |A| \cdot |\Omega|, H, 1/\epsilon, \log(1/\delta))$.*

*Proof.* The proof is similar to the approach in Section 3.1, which computes a near-optimal and 0-IC policy. We describe the differences below.

Instead of maintaining two-dimensional sets of inducible values, we split the dimension of the agent's value into two dimensions $v^{\mathsf{A}}$ and $v_*^{\mathsf{A}}$, which represent the agent's values under his truthful response (i.e., $\perp$) and his best deviation plan, respectively. Hence, each $\mathbf{v} \in \mathcal{V}(o)$ is now a tuple $(v^{\mathsf{P}}, v^{\mathsf{A}}, v_*^{\mathsf{A}})$.

(In Section 3.1, $v^{\mathsf{A}}$ and $v^{\mathsf{A}}_*$ are eventually forced to be the same, so there is no need to keep an additional dimension.)

The inducibility of a vector $\mathbf{v} = (v^{\mathsf{P}}, v^{\mathsf{A}}, v^{\mathsf{A}}_*)$ is characterized by the following constraints. First, we impose the same constraint as (3) on the first two dimensions of $\mathbf{v}$, so that they capture the players' payoffs under the agent's truthful response. In order for the third dimension $v^{\mathsf{A}}_*$ to capture the agent's maximum attainable value, we use a constraint similar to (4):

$$v^{\mathsf{A}}_* \geq \sum_{\omega^{\mathsf{A}}} p_{h-1}(\omega^{\mathsf{A}} \mid o) \max_{\widetilde{\omega}^{\mathsf{A}}} \sum_{a^{\mathsf{A}}} \max_{\tilde{a}^{\mathsf{A}}} \sum_{s, \omega^{\mathsf{P}}, a^{\mathsf{P}}} p_{h-1}(s, \omega^{\mathsf{P}} \mid o, \omega^{\mathsf{A}}) \cdot \bar{\pi}(\mathbf{a} \mid \omega^{\mathsf{P}}, \widetilde{\omega}^{\mathsf{A}}) \cdot$$
$$\left( r_h^{\mathsf{A}}\left(s, a^{\mathsf{P}}, \tilde{a}^{\mathsf{A}}\right) + v'^{\mathsf{A}}_*(s, \boldsymbol{\omega}, \widetilde{\omega}^{\mathsf{A}}, \mathbf{a}, \tilde{a}^{\mathsf{A}}) \right). \quad (8)$$

The remaining constraint is the same as (6).

All the non-linear constraints can be linearized the same way as the approach described in Section 3.1. Hence, we can efficiently approximate $\mathcal{V}_h(o)$ by examining the inducibility of points on a sufficiently fine-grained grid in $[0, H]^3$, which contains $\mathrm{poly}(H, 1/\epsilon)$ many points, and constructing the convex hull of these points. (Note that there is no need to ensure zero compromise on the agent's value as required in the proof of Lemma 4. This is because $\delta$-IC is defined with respect to the agent's expected value at the beginning of the game instead of that at every time step. Hence, using points on a grid suffices the purpose of the approximation in this proof.) The half-space representation of the convex hull can be computed efficiently given that it is in $\mathbb{R}^3$ [9]. Eventually, an optimal $\pi \in \widehat{\Pi}_\delta$ corresponds to a solution to $\max_{\mathbf{v} \in \mathcal{V}_1(\varnothing)} v^{\mathsf{P}}$ subject to $v^{\mathsf{A}} \geq v^{\mathsf{A}}_* - \delta$, and we can use the same forward construction procedure in Section 3.2 to compute $\pi_h(\cdot \mid \sigma; \omega^{\mathsf{P}}, \widetilde{\omega}^{\mathsf{A}})$.

Note that (8) only enforces $v^{\mathsf{A}}_*$ as an upper bound of the maximum attainable value, instead of the exact value. This suffices for our purpose because any $(v^{\mathsf{P}}, v^{\mathsf{A}}, v^{\mathsf{A}}_*)$ in the feasible set $\mathcal{V}_1(\varnothing) \cap \left\{ \mathbf{v} : v^{\mathsf{A}} \geq v^{\mathsf{A}}_* - \epsilon \right\}$ also implies the inclusion of $(v^{\mathsf{P}}, v^{\mathsf{A}}, \bar{v}^{\mathsf{A}}_*)$ in the same feasible set, where $\bar{v}^{\mathsf{A}}_*$ is the actual maximum attainable value induced by the policy that induces $(v^{\mathsf{P}}, v^{\mathsf{A}}, v^{\mathsf{A}}_*)$ according to our formulation. $\qquad\square$

**Theorem 10.** *There exists an algorithm that guarantees regret $\widetilde{\mathcal{O}}(\zeta^{1/3} T^{2/3})$ for both players with probability $1 - q$, where $\zeta = H^5 |S|^2 |A|^3 |\Omega|^2$. The computation involved in implementing the algorithm takes time $\mathrm{poly}(|S| \cdot |A| \cdot |\Omega|, H, T)$.*

*Proof.* We run reward-free exploration to obtain a model $\widehat{p}$ with error bound $\delta/2$. This can be achieved w.h.p. in $\widetilde{\mathcal{O}}(\zeta/\delta^2)$ episodes according to Lemma 7. Next, we compute an $\delta$-optimal strategy $\pi \in \widehat{\Pi}_\delta$ and use it in the remaining rounds. According to Lemma 9, this can be done in polynomial time.

By assumption, rewards are bounded in $[0, 1]$ so the regrets are at most 1 for both players in each of the exploration episodes. In each of the remaining episodes, the agent's regret is as follows, where we pick arbitrary $\rho^* \in \arg\max_\rho V_1^{\mathsf{A}, \pi, \rho}(\varnothing)$:

$$V_1^{\mathsf{A}, \pi, \rho^*}(\varnothing) - V_1^{\mathsf{A}, \pi, \perp}(\varnothing) \leq \underbrace{\left| \widehat{V}_1^{\mathsf{A}, \pi, \rho^*}(\varnothing) - \widehat{V}_1^{\mathsf{A}, \pi, \perp}(\varnothing) \right|}_{\leq \delta \text{ as } \pi \in \widehat{\Pi}_\delta} +$$
$$\underbrace{\left| \widehat{V}_1^{\mathsf{A}, \pi, \rho^*}(\varnothing) - V_1^{\mathsf{A}, \pi, \rho^*}(\varnothing) \right| + \left| \widehat{V}_1^{\mathsf{A}, \pi, \perp}(\varnothing) - V_1^{\mathsf{A}, \pi, \perp}(\varnothing) \right|}_{\leq \delta \text{ by Lemma 7}} \leq 2\delta.$$

The principal's regret is:

$$V^* - V_1^{\mathsf{P}, \pi, \perp}(\varnothing) = \max_{\pi' \in \Pi_0} V_1^{\mathsf{P}, \pi', \perp}(\varnothing) - V_1^{\mathsf{P}, \pi, \perp}(\varnothing)$$
$$\leq \underbrace{\max_{\pi' \in \widehat{\Pi}_\delta} V_1^{\mathsf{P}, \pi', \perp}(\varnothing) - V_1^{\mathsf{P}, \pi, \perp}(\varnothing)}_{\text{as } \Pi_0 \subseteq \widehat{\Pi}_\delta} \leq \underbrace{\max_{\pi' \in \widehat{\Pi}_\delta} \widehat{V}_1^{\mathsf{P}, \pi', \perp}(\varnothing) - \widehat{V}_1^{\mathsf{P}, \pi, \perp}(\varnothing) + \delta}_{\leq \delta \text{ as } \pi \text{ is } \delta\text{-optimal}} \leq 2\delta.$$

The reason that $\Pi_0 \subseteq \widehat{\Pi}_\delta$ is the following: Since $\widehat{p}$ ensures error bound $\delta/2$, we have $\left| \widehat{V}_1^{\mathsf{A},\pi',\rho}(\varnothing) - V_1^{\mathsf{A},\pi',\rho}(\varnothing) \right| \leq \delta/2$ for all $\rho$. By definition, $\pi' \in \Pi_0$ means that $V_1^{\mathsf{A},\pi',\perp}(\varnothing) \geq V_1^{\mathsf{A},\pi',\rho}(\varnothing)$. So, $\widehat{V}_1^{\mathsf{A},\pi',\perp}(\varnothing) \geq \widehat{V}_1^{\mathsf{A},\pi',\rho}(\varnothing) - \delta$ for all $\rho$; hence, $\pi' \in \widehat{\Pi}_\delta$.

The above bounds then lead to a total regret of at most $\widetilde{\mathcal{O}}(\zeta/\delta^2) + \mathcal{O}(T\delta)$ for each player. Taking $\delta = (\zeta/T)^{1/3}$ gives the upper bound $\widetilde{\mathcal{O}}(\zeta^{1/3}T^{2/3})$. $\qquad\square$

## B   Linearizing (3) and (4)

Specifically, to remove the maximization operator in (4), we introduce a set of auxiliary variables $y(a^{\mathsf{A}}, \omega^{\mathsf{A}}, \widetilde{\omega}^{\mathsf{A}})$ to capture the maximum values on the right hand side of (4). We replace the right hand side of (4) with $\sum_{a^{\mathsf{A}} \in A^{\mathsf{A}}} y(a^{\mathsf{A}}, \omega^{\mathsf{A}}, \widetilde{\omega}^{\mathsf{A}})$, and by adding the following constraint we force each $y(a^{\mathsf{A}}, \omega^{\mathsf{A}}, \widetilde{\omega}^{\mathsf{A}})$ to be an upper bound of the corresponding maximum value: for all $\tilde{a}^{\mathsf{A}} \in A^{\mathsf{A}}$,

$$y(a^{\mathsf{A}}, \omega^{\mathsf{A}}, \widetilde{\omega}^{\mathsf{A}}) \geq \sum_{s,\omega^{\mathsf{P}},a^{\mathsf{P}}} p_{h-1}(s, \omega^{\mathsf{P}} \,|\, o, \omega^{\mathsf{A}}) \cdot \bar{\pi}(\mathbf{a} \,|\, \omega^{\mathsf{P}}, \widetilde{\omega}^{\mathsf{A}}) \cdot \left( r_h^{\mathsf{A}}(s, a^{\mathsf{P}}, \tilde{a}^{\mathsf{A}}) + v'^{\mathsf{A}}(s, \boldsymbol{\omega}, \widetilde{\omega}^{\mathsf{A}}, \mathbf{a}, \tilde{a}^{\mathsf{A}}) \right)$$

$$(9)$$

To remove the quadratic terms in (3) and (9), we use an auxiliary variable $\mathbf{z}(s, \boldsymbol{\omega}, \widetilde{\omega}^{\mathsf{A}}, \mathbf{a}, \tilde{a}^{\mathsf{A}})$ to replace each term $\bar{\pi}(\mathbf{a} \,|\, \boldsymbol{\omega}) \cdot \mathbf{v}'(s, \boldsymbol{\omega}, \widetilde{\omega}^{\mathsf{A}}, \mathbf{a}, \tilde{a}^{\mathsf{A}})$ and impose the following constraint on $\mathbf{z}$:

$$\mathbf{H} \cdot \mathbf{z}(s, \boldsymbol{\omega}, \widetilde{\omega}^{\mathsf{A}}, \mathbf{a}, \tilde{a}^{\mathsf{A}}) \leq \bar{\pi}(\mathbf{a} \,|\, \boldsymbol{\omega}) \cdot \mathbf{b}, \qquad (10)$$

where $\mathbf{H}$ and $\mathbf{b}$ are taken from the half-space representation of the polytope $\widehat{\mathcal{V}}_{h+1}$, i.e., $\widehat{\mathcal{V}}_{h+1}(s, a^{\mathsf{P}}, \tilde{a}^{\mathsf{A}}) = \{\mathbf{x} : \mathbf{H} \cdot \mathbf{x} \leq \mathbf{b}\}$. It is straightforward that, when $\widehat{\mathcal{V}}_{h+1}(s, a^{\mathsf{P}}, \tilde{a}^{\mathsf{A}})$ is nonempty and bounded, (10) holds if and only if $\mathbf{z}(s, \boldsymbol{\omega}, \widetilde{\omega}^{\mathsf{A}}, \mathbf{a}, \tilde{a}^{\mathsf{A}}) = \bar{\pi}(\mathbf{a} \,|\, \boldsymbol{\omega}) \cdot \mathbf{x}$ for some $\mathbf{x} \in \widehat{\mathcal{V}}_{h+1}(s, a^{\mathsf{P}}, \tilde{a}^{\mathsf{A}})$.[5] Hence, (10) is the only constraint needed (for each tuple $(s, \boldsymbol{\omega}, \widetilde{\omega}^{\mathsf{A}}, \mathbf{a}, \tilde{a}^{\mathsf{A}})$) after we replace the terms with $\mathbf{z}$.

**The Complete Formulation**   In summary, the above approach yields the following linearized version of the constraint satisfiability problem in Section 3.1, where $\bar{\pi}$, $\mathbf{z} = (z^{\mathsf{A}}, z^{\mathsf{P}})$, and $y$ are the variables (highlighted in blue).

1. The value function constraint:
$$\mathbf{v} = \sum_{s,\boldsymbol{\omega},\mathbf{a}} p_{h-1}(s, \boldsymbol{\omega} \,|\, o) \cdot \left( \mathbf{r}_h(s, \mathbf{a}) \cdot \bar{\pi}(\mathbf{a} \,|\, \boldsymbol{\omega}) + \mathbf{z}(s, \boldsymbol{\omega}, \omega^{\mathsf{A}}, \mathbf{a}, a^{\mathsf{A}}) \right).$$

2. An IC constraint for each $\omega^{\mathsf{A}} \in \Omega^{\mathsf{A}}$:
$$\sum_{s,\omega^{\mathsf{P}},\mathbf{a}} p_{h-1}(s, \omega^{\mathsf{P}} \,|\, o, \omega^{\mathsf{A}}) \cdot \left( r_h^{\mathsf{A}}(s, \mathbf{a}) \cdot \bar{\pi}(\mathbf{a} \,|\, \boldsymbol{\omega}) + z^{\mathsf{A}}\left(s, \boldsymbol{\omega}, \omega^{\mathsf{A}}, \mathbf{a}, a^{\mathsf{A}}\right) \right) \geq \sum_{a^{\mathsf{A}} \in A^{\mathsf{A}}} y\left(a^{\mathsf{A}}, \omega^{\mathsf{A}}, \widetilde{\omega}^{\mathsf{A}}\right).$$

   Moreover, for each tuple $(a^{\mathsf{A}}, \omega^{\mathsf{A}}, \widetilde{\omega}^{\mathsf{A}}) \in A^{\mathsf{A}} \times \Omega^{\mathsf{A}} \times \Omega^{\mathsf{A}}$:
$$y\left(a^{\mathsf{A}}, \omega^{\mathsf{A}}, \widetilde{\omega}^{\mathsf{A}}\right) \geq \sum_{s,\omega^{\mathsf{P}},a^{\mathsf{P}}} p_{h-1}(s, \omega^{\mathsf{P}} \,|\, o, \omega^{\mathsf{A}}) \left( r_h^{\mathsf{A}}\left(s, a^{\mathsf{P}}, \tilde{a}^{\mathsf{A}}\right) \cdot \bar{\pi}(\mathbf{a} \,|\, \omega^{\mathsf{P}}, \widetilde{\omega}^{\mathsf{A}}) + z^{\mathsf{A}}\left(s, \boldsymbol{\omega}, \widetilde{\omega}^{\mathsf{A}}, \mathbf{a}, \tilde{a}^{\mathsf{A}}\right) \right).$$

3. An onward value constraint for each tuple $(s, \boldsymbol{\omega}, \widetilde{\omega}^{\mathsf{A}}, \mathbf{a}, \tilde{a}^{\mathsf{A}}) \in \bar{\Sigma}$:
$$\mathbf{H}\left(s, a^{\mathsf{P}}, \tilde{a}^{\mathsf{A}}\right) \cdot \mathbf{z}(s, \boldsymbol{\omega}, \widetilde{\omega}^{\mathsf{A}}, \mathbf{a}, \tilde{a}^{\mathsf{A}}) \leq \bar{\pi}(\mathbf{a} \,|\, \boldsymbol{\omega}) \cdot \mathbf{b}\left(s, a^{\mathsf{P}}, \tilde{a}^{\mathsf{A}}\right),$$

   where for every $o \in O$, the matrix $\mathbf{H}(o)$ and vector $\mathbf{b}(o)$ are given by the half-space representation of $\widehat{\mathcal{V}}_{h+1}(o)$, i.e., $\widehat{\mathcal{V}}_{h+1}(o) = \{\mathbf{v}' \in \mathbb{R} : \mathbf{H}(o) \cdot \mathbf{v}' \leq \mathbf{b}(o)\}$.

4. Additionally, we impose
$$\bar{\pi}(\cdot \,|\, \boldsymbol{\omega}) \in \Delta(A)$$

   for each $\boldsymbol{\omega} \in \Omega$ to ensure that $\bar{\pi}(\cdot \,|\, \boldsymbol{\omega})$ is a valid distribution over $A$.

---

[5]Note that if $\bar{\pi}(\mathbf{a} \,|\, \boldsymbol{\omega}) = 0$, then (10) imply that $\mathbf{z}(s, \boldsymbol{\omega}, \widetilde{\omega}^{\mathsf{A}}, \mathbf{a}, \tilde{a}^{\mathsf{A}}) = \mathbf{0}$: otherwise, the fact that $\mathbf{x}' = c \cdot \mathbf{z}(s, \boldsymbol{\omega}, \widetilde{\omega}^{\mathsf{A}}, \mathbf{a}, \tilde{a}^{\mathsf{A}}) + \mathbf{x}$ satisfies $\mathbf{H} \cdot \mathbf{x}' \leq \mathbf{b}$ for any $c \geq 0$ and $\mathbf{x} \in \widehat{\mathcal{V}}_{h+1}(s, a^{\mathsf{P}}, \tilde{a}^{\mathsf{A}})$ would prevent $\widehat{\mathcal{V}}_{h+1}(s, a^{\mathsf{P}}, \tilde{a}^{\mathsf{A}})$ from being bounded.

# C    Additional Discussion about Intractability without Hindsight Observability

The PSPACE-hardness can be seen by thinking of a POMDP as an instance of our problem where only the principal can make observations and perform actions to influence the environment (essentially, the agent can neither influence the principal nor the environment in this instance).

The PSPACE-hardness remains in the case of information design, where the principal observes the state directly but does not act, while the agent makes no observation but acts; as well as the case of mechanism design, where the agent observes the state directly but does not act, while the principal does not observe but acts. This can be seen by considering zero-sum instances, where the principal's and the agent's rewards sum to zero.

More specifically, consider for example the case of information design. If the goal is to compute the principal's maximum attainable payoff, the PSPACE-hardness of the problem is immediate: Since the game is zero-sum, it is optimal for the principal to not send no signal (if signaling were to improve the principal's payoff, the agent would be better-off just ignoring the signals). Hence, computing the maximum attainable payoff of the principal in this case is equivalent to computing (the negative of) the agent's maximum attainable payoff, which amounts to solving a POMDP.

One may argue that while the above example demonstrates the hardness of determining the principal's maximum attainable payoff, computing the principal's optimal policy is actually trivial in the example (i.e., sending no signal is optimal). So it does not rule out the possibility of an efficient algorithm which, given any sequence, computes the signal distribution of an optimal policy, without computing the principal's payoff the policy yields. It turns out that this is not possible, either.

Consider a game where the agent can choose between two actions $a$ and $b$ in the first time step. Action $a$ leads to a process where the principal's rewards are zero for all state-action pairs. Action $b$ leads to another process with payoffs $1 - x$ for the principal and $x$ for the agent, where $x \in [0, 1]$ depends on the principal's signaling strategy in this sub-process. For example, we can design this sub-process as a matching pennies game, where: nature flips a fair coin, the principal observes the outcome, and the agent must choose the same side of the coin to get a reward $1$ and otherwise he gets $-1$. If the agent plays this matching pennies game on his own, his expected payoff is $0$. The principal can reveal her observation to help the agent to improve the payoff. And the principal can do so probabilistically, so that she can fine tune the agent's expected payoff $x$ to any desired value in $[0, 1]$. To maximize the principal's payoff in the entire process requires finding an $x$ that is sufficiently high, so that the agent is incentivized to choose $b$ (otherwise, the principal only gets $0$); at the same time, we would like $x$ to be as low as possible to maximize the principal's payoff $1 - x$. This essentially requires knowing the agent's maximum attainable payoff in the sub-process following $a$, which is PSPACE-hard as we discussed above.

