# OpenReview forum: "Stochastic Principal-Agent Problems: Computing and Learning Optimal History-Dependent Policies"
_NeurIPS.cc/2025/Conference — NeurIPS 2025 poster_

### Official Review · Reviewer_bM1C · 2025-07-02

**Clarity:** 2
**Significance:** 2
**Originality:** 2
**Rating:** 4
**Confidence:** 2

**Summary:**

This work studies the problem of computing optimal history-dependent policies under a model of principal-agent style stochastic games. The problem is general-sum and the goal is to find a Stackelberg equilibrium among history-dependent strategies. This work makes a "hindsight observability" assumptions, where both players observe the full interaction history, including states, observations and actions at the end of each time step. Under this model, this work shows that it is possible to compute an $\epsilon$-optimal policy with time complexity scaling polynomially with the problem parameters. The key idea underlying this result is the construct of "inducible value sets", enabling the use of a value-based approach through sets that can be estimated with dynamic programming to compute convex polytope approximations of the value sets. This work also presents results on learning in such principal-agent problems, making use of recent advancements in reward-free exploration in RL.

**Questions:**

* In line 141-142, you say that the *hindsight observability* assumption is necessary, as without it the model subsumes POMDPs. Can you clarify what you mean here? The claim that it is necessary seems to imply that this is the *only* structural condition that can be made on top of what was described to make planning tractable. Is this the case? If not, perhaps this sentence should be clarified in the paper.
* You mention in the related work section that prior work [14] has shown optimal stationary strategies are inapproximable, leaving opening the tractability of optimal history-dependent policies. Is this work making the "hindsight observability" assumption? Wouldn't optimization over history-dependent policies be *harder*? Besides computational issues, a different motivation you discuss is the importance of history-dependence for optimality and capturing complex/interesting behaviors. Can you comment on the relationship between these two aspects?

**Ethical Concerns:**

["NO or VERY MINOR ethics concerns only"]

**Final Justification:**

I thank the authors for their responses and clarifications, which are helpful. In particular, they have addressed a concern I have about crediting related prior work (edited in review). However, since the main technical concerns have not been fully addressed, I will maintain my current score of a borderline accept, noting that I still lean towards acceptance.

**Limitations:**

The authors adequately addressed the limitations and potential negative societal impact of their work

**Quality:**

3

**Strengths And Weaknesses:**

**Strengths**

* This work identifies conditions under which it is possible to computationally-efficiently compute solutions to principal-agent-style stochastic games with partial information, which are known to be computationally-intractable in general.
* The importance of history-dependence in such principal-agent problems is well-motivated. The construct of inducible value sets effectively addresses the core challenge of the exponential growth of history-dependent policies.
* This work addresses both computation and learning, attaining polynomial complexity in both cases

**Weaknesses**
* The "hindsight observability" assumption appears quite restrictive---assuming that the states, observations, and actions are observable to both agents---given the motivations of the model as capturing aspects of imperfect and imbalanced information. However, the authors provide some discussion of this assumption in the main text and the appendix.
* The centralized learning assumption allows for the use of recent advancements in reward-free exploration in RL, but is perhaps not the best fit for the principal-agent style games in practice, where one would expect the two players to be learning in a decentralized manner with different observable information.
* ~The paper's discussion of related work contained some confusing phrasing that warrants revision. For example, [3] is described as 'concurrent work' (lines 114-115), despite being published two years ago. Also, I would suggest avoiding the phrase "while we discovered these techniques independently" in lines 110-111, which refers to work dating back to at least 2009.~
**Edit:** The authors have clarified this. Their preprint predates the mentioned works.

---

> ### Author Rebuttal · Authors · 2025-07-30
>
> Thank you for your very insightful and valuable comments and your positive assessment of our work!
>
> ---
>
> **Re: Necessity of hindsight observability**
>
> Thanks for this very insightful observation! You are right. The form of hindsight observability we assumed in the paper is not the only structural condition that could make the problem tractable. What we intended to mean here is that _some_ form of hindsight observability (not necessarily the one we used) is necessary due to the general intractability of POMDPs. We will clarify this to avoid confusions.
>
> ---
>
> **Re: Prior work [14]**
>
> Yes, [14] implicitly assumes hindsight observability. It considered a simpler model, where the principal observes everything about the environment, while the agent is missing part of the state that is modeled as an external parameter sampled based on an internal Markovian state. The external state is effectively hindsight observable in expectation. Our algorithm directly applies to the model of [14] and computes an optimal history-dependent policy.
>
> ---
>
> **Re: History-dependence**
>
> We understand that it may appear counterintuitive that it is hard to optimize over the space of history-dependent strategies—which is a larger space that also contains all stationary strategies. **However, when it comes to computational complexity, what matters is not the size of the feasible space, but rather how complicated its boundary is.** For example, an objective function might be easy to optimize over a convex space, but much harder over a subset of this convex space that is nonconvex.
>
> In our problem, while stationary stategies may appear to have a simpler form, **they actually impose additional constraints that force us to use the same strategy at the same state**. These additional constraints make the space of stationary strategies less well-shaped and hard to optimize over.
>
> ---
>
> **Re: Optimality of history-dependent strategies and complex bahavior they capture**
>
> Besides computational advantage, optimal history-dependent strategies in general also yield higher utilities than optimal stationary strategies. Roughtly speaking, consider for example a repeated version of the prisoner’s dilemma. The following tit-for-tat strategy can sustain mutual cooperation between the two players:
>
> - _Cooperate if the other player cooperated in the previous round; and defect, otherwise._
>
> The strategy is history-dependent because it depends on what the other player did in previous round. In contrast, no stationary strategy achieves this, as it effectively reduces the problem to a one-shot game, where defection is the dominant strategy for both players.
>
> P.S. We were a bit sketchy in the paper when it comes to these relations because we though they were elementary for the game theory community. However, we fully agree that elaborating on them in more detail would be very beneficial for the clarity of the paper, especially considering a broader audience.
>
> ---
>
> Please let us know if you have any further questions. We would be more than glad to provide additional clarifications.

---

> > ### Comment · Reviewer_bM1C · 2025-08-03
> >
> > Thank you to the authors for their response. While this clarification is helpful, it does not address the main technical concerns in my review, so I will maintain my current borderline rating.
> >
> > One aspect the authors have not commented on is their stated positioning within the literature, which I have found to be concerning in certain passages within the paper. For example, describing work from two years ago as "concurrent" (line 114). Typically, "concurrent" implies roughly simultaneous development, such as parallel submissions. I would encourage the authors to rephrase these sentences in the paper.

---

> ### Author Response · Authors · 2025-08-04
>
> Thank you so much for your very thoughtful follow-up comments! We greatly appreciate them. We just have a couple of additional points which we hope you could kindly consider.
>
> **We describe the work of Bernasconi et al. as concurrent because our preprint version predates both their preprint and their published version.** This timeline is supported by publicly archived records. We'd be happy to clarify this point further in the final version of our paper and include relevant context that is currently omitted due to anonymity requirements.
>
> We also realized that we overlooked another point you raised in the Strengths and Weaknesses section, so we add our response below. (We didn't mean to ignore it—we were just a bit overwhelmed as we had five reviews to address for this submission.)
>
> **Re: centralized learning:** Centralized learning can be viewed as a mechanism the principal commits to. Indeed, our mechanism ensures incentive compatibility in the sense that the agent's regret is sublinear. So, in this regard, the agent's behavior is incentivized rather than forced. This centralized learning paradigm aligns with other principal-agent-style mechinam design problems, which also involve some level of centralisms as the principal functions similarly to a central planner.
>
> Completely decoupling the two learners is an interesting direction. However, this is very challenging as our goal is effectively to find a Stackelberg equilibrium, which involves an optimization problem for the principal. This differs from the known convergence result of some decoupled learning dynamics to (coarse) correlated equilibria, as our goal is akin to reaching not only just an arbitrary equilibrium but also an optimal one. To the best of our knowledge, even in the much simpler setting of repeated normal-form Stackelberg games, there is currently no elegant approach in the literature to steer a completely independent learning agent into a Stackelberg equilibrium. For this reason, we were unable to pursue this line of work further in the present paper.
>
> ---
>
> Please let us know if you have any further questions!

---

> > ### Comment · Reviewer_bM1C · 2025-08-04
> >
> > Thanks to the authors for the clarification on the point regarding related work--that clears things up. Thank you also for the discussion of the challenges of decentralized learning. I will maintain my concern, continuing to lean towards acceptance, and will keep an eye out for any discussion among the AC and reviewers post-rebuttal.

---

> > > ### Author Response · Authors · 2025-08-05
> > >
> > > Thank you! We greatly appreciate your time in the discussion as well as your very thoughtful feedback.

---

### Official Review · Reviewer_o9Qq · 2025-07-02

**Clarity:** 3
**Significance:** 3
**Originality:** 3
**Rating:** 4
**Confidence:** 3

**Summary:**

This paper considers a stochastic principal agent problem with history dependent policies. Once this generalized principal agent problem is introduced, the paper makes the following main contributions:

1. It is shown that there exists an approximately optimal incentive compatible (IC) policy and the paper proposes a polynomial time procedure to compute the induced distribution for any given history of interaction and observation tuple in polynomial time (Theorem 5). From the technical viewpoint the procedure requires to compute polytope approximations of the so-called inducible value sets introduced by the paper using dynamic programming before one can unroll the optimal policy for a given history and observation tuple.

2. In the episodic online learning setting where the transition model is unknown for the players, the paper proposes a centralized learning algorithm which guarantees a regret of order $T^{2/3}$ to both players with high probability, and which can be executed in polynomial time in the problem parameters (Theorem 10). The algorithm follows a explore-and-commit strategy which involves a first reward-free exploration step to learn an estimate of the transition kernel followed by a second step for computing an approximately optimal approximate IC policy using the estimated kernel. In particular, the learning algorithm requires relaxing the IC constraints compared to the result of the previous section.

**Questions:**

1. Clarification question about hindsight observability: Both players (and the agent in particular) observe the full history *at the end* of each time step. I find it a little confusing that the deviation plan (l. 176) for the agent takes the history as input (including the last principal’s observation and action?). Can you clarify this?

2. Before committing to approximation, is there anything known or that can be said and exploited about the geometry of exact value sets beyond nonlinearity? Also can you make the point in l. 248 clearer? Ou argue about nonlinearity: What are the quadratic terms in the expressions? Also there is no max term in (3) which seems linear in v’. Nonlinearity is somehow clear from the max but less so from the rest.

3. What is exactly the output in step 3 of Fig. 2? You say $\bar{\pi}$ (which is a one-step policy as mentioned in l. 225), is it the last one computed in the loop? Is the output the ‘concatenation’ of the $h-1$ ones computed so that it’s a history dependent policy? It’s a little confusing.


4. Is there a way to compute an optimal policy recursively without needing to compute first all the inducible value sets and then perform the unrolling? Say simultaneously using the inducible value sets and approximate the optimal policy for a given history and observation tuple if that makes sense.

5. l. 341-348: I am a little confused by the comparator in this learning part. You resort to relaxation for IC constraints which seems reasonable, but then the final comparator in the regret is not adjusted for this, i.e. you still compare to V^* which is the optimal value of (2) where the lower level (agent’s problem) is not relaxed with $\delta$-IC.
Can you clarify why you can still achieve vanishing (average) regret w.r.t $V^*$ and the (true) best response for the agent? Do you reduce $\delta$ in some way? Otherwise there should still be a bias in the final regret bound and a dependence on $\delta$ I guess.

6. Is there a way to provide a sharper characterization of the dependence on $\epsilon$ for the problem, more than just polynomial? Is it related to the discretization? Can such a discretization be avoided?

**Additional, more minor questions:**

7. Optimal policy unrolling: This kind of procedure reminds me of what is sometimes called *local planning* in RL which can be done efficiently in polynomial time given a state for instance to compute the optimal policy at that state (see e.g. Szepesvari’s lecture notes online). Is this accurate?

8. Is the order of regret a consequence of the reward free exploration procedure? Because the quality of the approximation (order in epsilon) is not really specified in Theorem 5 whereas the order is precisely given in Theorem 10. Is it optimal up to improving the reward free exploration step? Is there still room for improvement?

**Ethical Concerns:**

["NO or VERY MINOR ethics concerns only"]

**Final Justification:**

The rebuttal response addresses my concerns regarding generality of the PA model and the positioning of the work with respect to prior work, computation of the optimal policy and the hindsight observability feedback assumption. The illustrative examples for this assumption are appreciated. I believe that the ability to enable history dependence in optimal policy design is a nice feature of this work within the growing literature of principal agent problems. I maintain my positive attitude towards this paper as per my initial review and recommend acceptance.

**Limitations:**

See weaknesses section.

**Paper Formatting Concerns:**

Nothing to mention.

**Quality:**

3

**Strengths And Weaknesses:**

**Strengths**

- Solving repeated principal agent problems is an active topic of research nowadays, including in the ML community. The paper adds to this literature by considering a sequential stateful setting with far-sighted agents and history dependent policies beyond simpler stationary policies. In my opinion, this is an interesting addition although I am not an expert in principal-agent problems and I leave experts of the room provide a more accurate judgment in view of existing literature. Considering history dependent policies definitely brings some complexity to the problem which is balanced by the hindsight (full) observability assumption. The appendix and technical developments (which I did not go over in details) seem rather compact.

- The paper is overall well-written, notations are precise and the paper is generally easy to follow. Although the mathematical objects involve some complexity, the paper did a good job introducing nice and rigorous notations which make the paper readable, especially for (recursive) induced value sets, dynamic programming procedures, summaries of the algorithms in frames and a precise introduction of the problem in section 2 including the communication protocol and the setting of interest with the required assumptions.

- The results are sound but I have not checked the proofs in the appendix.

**Weaknesses**

- One of the highlighted contributions is a **generalized** framework for principal-agent problems. With this regard, I would expect a clearer positioning of the paper w.r.t the literature. The paper discusses related work. Nevertheless, it is not immediately clear to me from the paper which principal-agent problems are particular cases of the setting of this paper in the literature, among the papers dealing with sequential **stateful** settings (MDPs) which are the most relevant to this paper (if there are any). I think that would clarify further the generality of the model and the positioning of the paper:

-- More precisely, can you elaborate more precisely on the differences than in l. 95-108? The model seems different in several aspects via communication and the use of history dependent policies for instance as well as the setting itself (not contract design like in Wu et al. 24, Ivanov et al. 24 and others) …

-- Can you also clarify the positioning w.r.t. Stackelberg (Markov) games? E.g. Bai et al. Neurips 2021 Sample-Eﬃcient Learning of Stackelberg Equilibria in General-Sum Games. There are also several more recent works about learning Stackelberg equilibria.

- l. 317-323: About hindsight observability and reward-free learning: Are the sampling requirements aligned here? You reframe your problem as an MDP but do the MDP works you use rely on a generative model (which is a standard assumption in RL), i.e. the ability to query a next state from any given state action (not necessarily only the ones observed online during the interaction)? Is this a reasonable assumption in your principal-agent setting with hindsight observability? Is it reasonable to assume that one can query the next state from any state $\theta$ (which combines actions of both players for instance) while hindsight observability seems to only give access to a history of interactions between the principal and the agent?

- One limitation might be that the algorithm is centralized. I would imagine that players in a Principal-Agent setting would rather act independently. Who is computing the optimal policy exactly? The principal? Is it reasonable to assume the principal (or some central entity) has access to all information (in hindsight) for both players (actions for instance)?

 **Typos and suggestions**

- l. 234: is it rather $\tilde{\omega}^A$ here for the IC constraint?
- It might be useful to remind in Fig. 2 step 1 that $v= (v^P, v^A)$ and that $v^P \in \mathbb{R}$.
- l. 340: I guess $\delta$ should be $\epsilon$ here?

---

> ### Author Rebuttal · Authors · 2025-07-30
>
> Thanks a lot for the very thoughtful comments! We find them very valuable for improving our paper. We will cerntainly fix the citations and other minor issues; thanks for pointing them out.
>
> Our answers to your other questions are as follows. If anything remains unclear, we would be more than glad to provide further clarifications.
>
> ---
>
> First, regarding your questions in the **Strengths and Weaknesses** section:
>
>
> 1. **Generative model.** To the best of our knowledge, the reward-free exploration algorithm of Jin et al. (2020) does not rely on a generative model. The algorithm relies only on state-action pairs encountered online. They have in particular introduced associated concepts such as $\delta$-significant states to handle states that are hard to reach. Therefore, we believe that not having access to a generative model would not be an issue for our algorithm. (But please let us know if we didn’t understand your question correctly.)
>
>
> 2. Our algorithm is **not** fully centralized in the sense that the principal does **not** have any control over the agent, who are free to deviate to whatever actions they prefer. Hindsight observability is indeed the case in many models such as games with full observability (which is fairly common in the literature) and information design (where one of the players have full observability). (Please also see our discussion at Lines 145--149.) From a computational complexity point of view, some degree of hindsight observability is needed for the setting to be tractable; otherwise, the problem would be as hard as POMDPs, which are PSPACE-hard.
>
> ---
>
> Next, regarding your questions in the **Questions** section:
>
> **Re: Q1**
>
> In the deviation plan, the agents takes into account the history in the previous time steps, which are seen by the agent because of hindsight observability. **In the current  time step, the agent only has access to their own observation**, so as we described at Line 176, the deviation plan does **not** rely on the principal’s observation in the current time step.
>
> **Re: Q2**
>
> In Equations (3) and (4), **the quadratic terms are the products of the $\bar{\pi}$’s and $v$’s (i.e., terms highlighted in blue in the paper)**. When we say “the quadratic terms and the maximization operator in (3) and (4)”, we mean the quadratic terms in both (3) and (4) and the maximization operator only in (3); we do **not** imply that each equation contains both types of elements.
>
>
> We did not find any other properties of the value sets that can be used to design efficient approaches to computing the exat value sets. All we know is that each set is a convex polytope defined by possibly exponentially many hyperplanes.
>
> **Re: Q3**
>
> In Line 3 of Figure 2, the value of the output is equal to $\bar{\pi} ( \cdot | \tilde{\omega}^P, \omega^A)$, while it is interpreted as the distribution $\pi( \cdot | \sigma; \tilde{\omega}^P, \omega^A)$ (which is what the algorithm is expected to compute). Perhaps this line would be easier to understand if we break it into the following two lines:
>
>
> -  First, assign the value of $\bar{\pi} ( \cdot | \tilde{\omega}^P, \omega^A)$ to $\pi( \cdot | \sigma; \tilde{\omega}^P, \omega^A)$;
>
>
> - Then, return $\pi( \cdot | \sigma; \tilde{\omega}^P, \omega^A)$.
>
>
> And, yes, you are right: the value of $\bar{\pi}$ in Line 3 is computed in the last iteration of the for-loop in Line 2.
>
>
> Note that (as explained in the first paragraph of Section 3.2) our goal is to compute the action distribution defined by $\pi$ for any given history, rather than the explicit representation of $\pi$. And $\pi$ is history-dependent because for different input sequences, the algorithm in Figure 2 will output different distributions. So $\pi$ prescribes different joint action distribution for different histories.
>
> **Re: Q4**
>
> This is a very interesting question. We don’t find any way to compute the policy and the inducible value sets simultenously. One possibility is to use policy-based approaches that compute optimal policies without maintaining the value sets. For example, if we could compute the gradient of the value function w.r.t. the policy, we might be able to use gradient descent to optimize the policy directly. This is a very interesting direction, but we have little understanding so far. Computing the gradient appears very challenging especially because we must also avoid using explicit representations of history-dependent policies at the same time. To work with the gradient of something that does not even have an explicit representation---if possible at all---appears to require highly innovative techniques.
>
> **Re: Q5**
>
> Yes, you are right, when it comes to the regret, we still compare the value achieved against the optimal value achivable under the exact IC constraint. This is actually an easier benchmark than the one with the $\delta$ relaxation. To see this please note that any solution to the problem with the exact IC constraint is also a feasible solution to the one with the $\delta$ relaxation.
>
> **Re: Q6**
>
> We are not aware of any approach that can further reduce the polynomial dependence on $1/\epsilon$ in the time complexity. An associated open question is whether we can compute the exact solution efficiently. The latter has been shown to be possible in the special case of turn-taking games, where only one player acts in each time step and there is no information asymmetry. It remains unknown though whether this is also possible in our much more general model where the players move simultaneously under information asymmetry.
>
> **Re: Q7**
>
> Yes, there are some similarities between our approach and local planning in RL you mentioned. Both of them are recursive implementation and are based on the Bellman equation (or its variant). But the unrolling process in each iteration is a lot more complicated in our case. It requires solving the problem we described in Section 3.1, instead of just a maximization problem as in RL local planning.
>
> **Re: Q8**
>
> Yes, you are right, the regret bound is a consequence of reward-free exploration. There is no $\epsilon$ or $\delta$ terms in the time complexity in Theorem 10 becasue we use specific $\epsilon$ and $\delta$ that are polynomials of the other terms to derive the regret bound. The details can be found in the proof of Theorem 10 in the appendix. It could be that the bound is tight but we don’t have any evidence so far; a lower regret bound can help to resolve this question.
>
> ---
>
> Please let us know if anything remains unclear. We’d be very glad to provide more clarifications.

---

> > ### Comment · Reviewer_o9Qq · 2025-08-08
> > **Rebuttal acknowledgement**
> >
> > I would like to thank the authors for their response which addresses most of my concerns and questions. I apologize for the delay in my response but I do have two follow-up questions regarding my previous questions that I think were not addressed precisely:
> >
> > - Regarding the generality of the PA framework proposed (see my first point in weaknesses), is this generality coming from an immediate 'generalization' of existing general PA frameworks in the literature to the sequential setting? Which works in the sequential (stateful) setting are subsumed by this work?
> >
> > - Regarding the algorithm, as the authors mention in the paper (l.292), it is centralized. I understand that the principal does not have any control over the agent as mentioned by the authors' response. As per my initial review, I still would imagine that players in a Principal-Agent setting would rather act independently (i.e. each one computing their strategy using the information they have access to). Who is computing the optimal policy exactly? The principal or a central entity having access to all interaction information to be able to compute all inducible sets and the unrolling procedure? Is it reasonable to assume that some central entity has access to all information (in hindsight) for both players (actions for instance)? I guess some elements of response were provided as a response to reviewer bM1C that I had a look at but I would appreciate further precisions here regarding the questions above.

---

> > > ### Author Response · Authors · 2025-08-08
> > >
> > > Thanks a lot for taking time to assess our responses and for posing the very insightful follow-up questions!
> > >
> > > **Re: Positioning of our work**
> > >
> > > Our model is a sequential generalisation of Myerson's (1982) classic one-shot principal-agent model. The most relevant sequential stateful settings it encompasses include several works on information design (aka, Bayesian persuasion) [Gan et al. 2022; Wu et al. 2022; Bernasconi et al. 2022, 2023] (which is the special case where the principal observes everything about the environment while the agent only observes an internal Markovian state). In the case without information asymmetry, our model becomes stateful Stakcelberg games with coordination [Letchford and Conitzer, 2010; Letchford et al. 2012] There are also more recent models on MDP-based sequential contract design [Wu et al. 2024; Ivanov et al. 2024]. With small tweaks to incorporate payments, our algorithm can compute optimal history-dependent strategies in these models as well.  We will update our related work section accordingly to clarify the positioning of our work.
> > >
> > > **Re: Who is computing the optimal policy?**
> > >
> > > In our problem, it is the principal who is computing the optimal policy (or equivalently, we compute the optimal policy on behalf of the principal). At a high level, this can be viewed as a Stackelberg game, where the principal acts as the leader and the agent acts as the follower. In equilibria of the game, the principal commits optimally, under the assumption that the agent will always play a best response to the committed strategy.
> > >
> > > **Re: Assumption on information access**
> > >
> > > In real-world scenarios, it is fairly common for players to observe each other's actions after they are played (e.g., playing rock-paper-scissors repeatedly). Besides actions, in many settings, players are regularly updated about each other's private observations, too. This can happen in particular when the state comprises multiple components, with each player having access to only a subset of the components. When the subset each player has access to is common knowledge, all players’ private observations effectively become public when the state is revealed at the end of each time step. Some concrete examples are as follows:
> > >
> > > - Traders may each observe local or regional information about the market, which are in some cases aggregated later and released publicly. In energy market, for example, regional grid operators obtain first-hand information about the supply and demand in their own regions (but not in other regions), and later receives system-wide reports covering the entire market.
> > >
> > > - In an R&D consortium, each firm privately evaluates prototypes or test results and acts on that information during the reporting cycle. At the cycle’s end, all such results are disclosed to consortium members as per agreed rules.
> > >
> > > As we mentioned in our previous response, from a complexity-theoretic perspective, the hindsight-observability assumption is an inevitable compromise due to the intrinsic PSPACE-hard of POMDPs. Without hindsight observability, efficient algorithms _that compute optimal solutions_ are hopeless. We can still apply our algorithm to the problem and the algorithm will generate some policy in polynomial time. But this policy is not necessarily optimal unless hindsight observability holds. That said, since our algorithm enables history-dependence, the policy it generates is guaranteed to outperform the best _stationary_ policy (which maps solely from the principal's current observation), regardless of whether hindsight observability holds.
> > >
> > > We will incorporate these responses in the paper. Thanks again for these insightful questions and please let us know if anything remains unclear!
> > >
> > > ---
> > >
> > > **References**
> > >
> > > - R B Myerson. Optimal coordination mechanisms in generalized principal–agent problems. Journal of mathematical economics, 10(1):67–81, 1982.
> > > - J Gan, R Majumdar, G Radanovic, and A Singla. Bayesian persuasion in sequential decision-making. AAAI ’22
> > > - J Wu, Z Zhang, Z Feng, Z Wang, Z Yang, M I. Jordan, and H Xu. Sequential information design: Markov persuasion process and its efficient reinforcement learning. EC ’22.
> > > - M Bernasconi, M Castiglioni, A Marchesi, N Gatti, and F Trovò. Sequential information design: Learning to persuade in the dark. NeurIPS ’22
> > > - M Bernasconi, M Castiglioni, A Marchesi, and M Mutti. Persuading farsighted receivers in MDPs: the power of honesty. NeurIPS ’23
> > > - J Letchford and V Conitzer. Computing optimal strategies to commit to in extensive-form games. EC ’10.
> > > - J Letchford, L MacDermed, V Conitzer, R Parr, and C L. Isbell. Computing optimal strategies to commit to in stochastic games. AAAI '12.
> > > - J Wu, S Chen, M Wang, H Wang, and H Xu. Contractual reinforcement learning: Pulling arms with invisible hands, 2024. https://arxiv.org/abs/4742407.01458.
> > > - D Ivanov, P Dütting, I Talgam-Cohen, T Wang, and D C. Parkes. Principal-agent reinforcement learning, 2024. https://arxiv.org/abs/2407.18074.

---

> > > > ### Comment · Reviewer_o9Qq · 2025-08-09
> > > > **Thank you**
> > > >
> > > > I would like to thank the authors for their detailed response which addresses my concerns regarding generality of the PA model, computation of the optimal policy and the feedback setting. The examples are appreciated. I do not have any further questions and I maintain my positive attitude towards this paper.

---

### Official Review · Reviewer_HYK4 · 2025-07-02

**Clarity:** 3
**Significance:** 3
**Originality:** 3
**Rating:** 5
**Confidence:** 4

**Summary:**

The paper shows that optimal and exactly IC solutions can be computed efficiently in two-player stochastic games with asymmetric information by extending a recent idea of using value polytopes.

**Questions:**

Could you please comment on my questions above? Specifically:

1. Technical obstacles
2. Assessment of the difference with Gan et al. and Bernasconi et al.
3. Intuition for point 3 of figure 1.

**Ethical Concerns:**

["NO or VERY MINOR ethics concerns only"]

**Final Justification:**

The authors did a great job at answering my questions. I now have a better appreciation of the contribution, and have raised my score accordingly.

As discussed below, the authors resovled my questions regarding the connection with existing work, and have also resolved a concern on a technical claim.

**Limitations:**

Yes

**Paper Formatting Concerns:**

No concerns

**Quality:**

3

**Strengths And Weaknesses:**

Computation of Stackelberg equilibria in stochastic games is not my area of expertise, so I will focus on clarity of the contribution.

Overall, I find the paper well written. The technique used in the paper, based on value polytopes, seems to have been used already in a few papers (as mentioned by the authors at the end of the related work section). However, if I understood correctly, the authors claim that they are able to use the idea of inducible values to guarantee exact incentive compatibility, unlike prior work. So, there seems to be a tangible improvement, though it is hard for me to quantify how technically challenging (or, conversely, incremental) this is. I would have appreciated more clarity regarding the technical obstacles that the authors had to overcome to establish this refined result.

Another point of comparison is the paper by Gan et al.. That paper is for turn-taking games, whereas the current paper seems to be more general, although somewhat related in terms of techniques (Gan et al. also keeps a sort of "Pareto frontier" via constraints if I understood correctly). Is this assessment accurate?

Overall, I found the discussion in Section 3 quite terse. An example could have gone a long way. Furthermore, I am still slightly puzzled about how and why it is possible to compute the H-representation of the value polytope efficiently, given that usually converting from V- to H-polytopes (say, using double description techniques) cannot be tractable in general. Is there any intuition regarding this?


Minor points:
- Citations [2], [6], [17], and [40] are to an arXiv preprint. Were the corresponding papers published anywhere? If so, please update to use the published version, unless there is a specific reason not to.
- Citations [3] and [22]: "pomdp" and "mdp" should be "POMDP" and "MDP", respectively.
- Citation [34]: "markov" -> "Markov"
- Citation [27]: the citation is missing a venue, journal, publisher, or similar.

---

> ### Author Rebuttal · Authors · 2025-07-30
>
> Thank you for your very insightful comments and positive assessment of our work! We will cerntainly fix the citations and other minor issues you pointed out. Thanks a lot for pointing them out.
>
> ---
>
> **Re: Difference with Gan et al., and Bernasconi et al. (as well as Zhang et al.)**
>
> _(It seems that in your comments the paper you wanted to refer to is Zhang et al. 2023—titled “Efficiently solving turn-taking stochastic games with extensive-form correlation”—but you pointed out Gan et al. instead. Please see our remark about the paper of Zhang et al. at the end of this answer.)_
>
> Gan et al. (2022) first introduced a similar sequential information design model, **but their focus is on stationary strategies, which is less general than history-dependent strategies we considered in our paper**. Gan et al. showed the intractablity of stationary strategies (NP-hard to optimize even approximately), which is one of the main motivations behind our consideration of history-dependent strategies. To the best of our knowledge, Gan et al.’s approach did not involve constructing inducible value sets or anything alike. Their main positive result is an efficient algoritm that works but only with myopic (or advice-myopic) agents. In our paper (and that of Bernasconi et al.), the agent is far-sighted (which is strictly more general than myopic agents).
>
> **Our work is in fact concurrent with Bernasconi et al.** (their preprint version appeared after ours but their work got published ealier). We would therefore hope that our contributions could be evaluated based on their own merits, rather than being considered as subsequent to Bernasconi et al. **Besides preceeding the work of Bernasconi et al., our results are also stronger**, as we explain below:
>
> -  Our algorithm provides better guarantees: using a novel slice-based discretization, it ensures **exact IC (incentive compatibility)**. In comparision, the approach of Bernasconi et al. only ensures **approximate IC**, whereby the IC constraint is alway violated by a $\delta > 0$ amount.
>
> - Our model allows partial observability of both the principal and the agent, and two-way communication between them. In comparision, **Bernasconi et al. considered an information design model, which is only a special case of ours.** In their model, only the agent has partial observability and communication is uni-directional: signals only go from the principal to the agent, not the other way round.
>
> - In addition to the computation problem, we also explored the learning version of the problem in Section 4, whereas this was not considered by Bernasconi et al.
>
> P.S. It seems that in your comments the paper you wanted to refer to is Zhang et al. 2023, instead of Gan et al. 2022. In the paper of Zhang et al., they considered turn-taking games and used the notion of Pareto frontier. The high level idea of their approach and ours are similar, but since they are consider a more special case of the model (that is turn-taking and without any information asymmetry), they were able to design a very sophasiticated algorithm to compute an exact optimal solution, without involving any approximation errors. It remains open whether this approach can be extended to our more general setting (which appears very challenging, perhaps not even possible, as the approach is already highly sophasiticated even with turn-taking games.)
>
> ---
>
> **Re: Technical obstables**
>
> Following our responses above, the main technical obstables we overcome in this paper are the following:
>
> 1. Enabling the use and efficient computation of history-dependent strategies (which were also addressed concurrently by Bernasconi et al.). This is non-trivial especially because of the exponential growth of history trajectories.
>
> 2. A novel slice-based discretization approach to ensure exact IC (which is missing in the work of Bernasconi et al.).
>
> 3. Application of reward-free exploration to the learning version of the problem to achieve efficient learning.
>
> ---
>
> **Re: Intuition of Point 3, Figure 1**
>
> The Half-space representation can be computated efficiently because the polytope is in $\mathbb{R}^2$. In fact, an easy algorithm that builds the H-representation of the convex hull of a set $P$ of points works as follows: enumerate ever pair of points in $P$, compute the line passing through each pair, and check if all points in $P$ are on the same side of this line (the same half-space). If yes, then keep this line as a linear constraint in the H-representation; otherwise, discard it.
>
> This can further be generalized to **any fixed dimension $d$**, where we enumerate every combination of  $d$ points in $P$ and compute the hyperplane passing through these points. There are only polynomially many such combinations when $d$ is fixed.
>
> P.S. If I understand correctly, the conversion between V- and H-representations of a polytope is hard only when the number of dimensions is **not** fixed. This corresponds to the setting where the number of agents is not fixed in our model. As we also mentioned in the response to reviewer RCDp, if the number of agents is not fixed, then our problem (which subsumes computing an optimal correlated equilibrium) would be NP-hard even when the game is one-shot (as implied by [Papadimitriou & Roughgarden, 2008]).
>
> ---
>
> **Reference:**
>
> - Christos H Papadimitriou and Tim Roughgarden. Computing equilibria in multi-player games. SODA ‘05.
>
> ---
>
> Please let us know if you have any further questions. We would be more than glad to provide additional clarifications.

---

> > ### Comment · Reviewer_HYK4 · 2025-08-07
> >
> > Dear authors, thank you for your response.
> >
> > You are right. In my review, I had a comment where I wrote Gan et al. when I instead meant Zhang et al. Thank you for calling that out.
> >
> > About the H- vs V-polytope discussion: you are right, I was thinking of the dimension as free rather than it being a constant in my question. Your rebuttal settles my question.
> >
> > I really appreciated the detailed response. I agree with your reconstruction of the timelines and different contributions.
> > I will raise my already positive score to signal increased appreciation for the paper.

---

> > > ### Author Response · Authors · 2025-08-07
> > >
> > > Thanks a lot for taking the time to review our responses. We truly appreciate your thoughtful assessment and feedback.

---

### Official Review · Reviewer_o1dQ · 2025-07-03

**Clarity:** 2
**Significance:** 3
**Originality:** 3
**Rating:** 4
**Confidence:** 3

**Summary:**

The paper studies a dynamic principal-agent problem modeled by POMDP, as the policy could depend on the history. It aims to find a policy for the principal so as to maximize its reward, assuming that the agent responds optimally. Under hindsight observability, the paper proposes an algorithm based on computing inducible value sets and optimal policies. Then the paper considers an online setting with episode RL.

**Questions:**

Motivation: Why is it necessary to consider POMDP instead of a normal MDP?

Since the policy is history-dependent, there could be a scenario tree getting involved in the algorithm, but how the authors address the curse of dimension remains unclear.

Contribution in Line 62, it should be stated what the key motivation is for addressing history dependence.

The notation system is very hard to understand. For example, there are multiple notations at the beginning of Sec 2 and a lot of new notations in Sec 2.1. In addition, the same notation is widely abused with several meanings and several different input set.

Thm 3.7, what is the definition of  an $\epsilon$-optimal IC policy?

**Ethical Concerns:**

["NO or VERY MINOR ethics concerns only"]

**Final Justification:**

The paper studies a challenging problem and addresses it well. The writing is an issue yet I believe that authors could handle it properly.

**Quality:**

2

**Strengths And Weaknesses:**

The idea to overcome NP hardness via hindsight observability and computing the optimal policy in polynomial time is quite surprising.

The major weakness is that the writing makes it very hard to comprehend the main idea and verify the claims made in the paper.

A few questions related to motivation: why it is important to consider history-dependent policy instead of a Markov policy?

Despite the fact that the paper seems to achieve a poly time algorithm for the very hard POMDP problem, the lack of experiments in the main context makes it hard to judge how significant the improvement it achieves.

---

> ### Author Rebuttal · Authors · 2025-07-30
>
> Thanks a lot for your very insightful comments and questions; they are greatly appreciated! Below are our answers to the your questions.
>
> ---
>
> **Re: Why POMDP instead of MDP?**
>
> We consider POMDP because it captures information asymetry between different players, while a standard MDP is insuffient for this purpose.
>
> ---
>
> **Re: Why history-dependence?**
>
> We introduce history-dependence to the model and view it as the main contribution of the work because of the following reason (as we also explained between Lines 49 to 69 on Page 2):
>
> 1. **History-dependent strategies offer higher utility than stationary (Markovian) ones.** History-dependent strategies are by definition a generalization of stationary ones. In the multi-player setting we consdier, they offer strictly higher utilities than stationary strategies in many problem instances. (For example, tit-for-tat is a history-dependent strategy, which is necessary for achieving optimality in some sequential games.) This is very different from (single-player) MDPs, where stationary strategies are always sufficient for achieiving optimality.
>
> 2. **History-dependent strategies are computationally more advantagous than stationary (Markovian) ones.** Besides being suboptimal, stationary strategies are also computationally intractable, as reveald by previous work [Gan et al., 2022]—it is NP-hard even just to approximate the best stationary strategy. In contrast, our main finding in this paper is that optimal history-dependent policies can be computed in polynomial time (in the same setting where stationary strategies are intractable).
>
> ---
>
> **Re: How did we address the challenges of handling history-dependent strategies**
>
> Yes, you are right. History-dependent strategies are hard to deal with because of the exponential growth of possible histories as the time horizon gets longer (if we understand correctly, this is what you meant by "scenario tree" in your comments). For each history, one action distributon needs to be defined, so explicit representations of history-dependent strategies would be too costly to be efficiently managable. Our approach can be viewed as a **compact representation** to overcome this representational issue. Effectively, the algorithm in Figure 2, along with the inducible value sets computed by the algorithm in Figure 1, encodes a history-dependent policy $\pi$, so that for any given history trajectory $(\sigma;\omega^P, \tilde{\omega}^A)$ (i.e., any path in the scenario tree), the algorithm computes the action distribution $\pi(\cdot | \sigma;\omega^P, \tilde{\omega}^A)$ defined by this policy. This is a key technical novelty our work introduced to overcome the challenges of handling history-dependent strategies. Please also refer to our discussion in Section 3.2 for more details.
>
> ---
>
> **Re: Definition of $\epsilon$-optimal IC policy**
>
> An $\epsilon$-optimal IC policy is a policy that is $\epsilon$-optimal and also IC. IC is defined in Defintion 1 (Page 5). And $\epsilon$-optimality means that the payoff it offers is as high as that offered by an optimal policy, up to a difference of at most $\epsilon$ (as explained in Lines 186--187, Page 5).
>
> ---
>
> **Reference:**
>
> - Jiarui Gan, Rupak Majumdar, Goran Radanovic, and Adish Singla. Bayesian persuasion in sequential decision-making. AAAI ‘22.
>
> ---
>
> We sincerely hope that our answers above have clarified your questions and would be very grateful if you could consider raising your scores accordingly. If anything remains unclear, please let us know and we would be more than glad to provide further details!

---

> > ### Comment · Reviewer_o1dQ · 2025-08-01
> >
> > After carefully checking Sec 3.2, I am still not fully convinced by the response about challenges of handling history-dependent strategies, i.e., how to avoid exponential dependence on the length of the history. It seems that the proposed technique does not sacrifice anything to achieve this goal using the so-called compact representation.
> >
> > Figure 1 says one needs to use dynamic programming. Given the random nature of POMDP, conducting dynamic programming would lead to exponential dependence on the length of the history, unless one solves the deterministic problem under the full information setting? Could you please clarify? Figure 2 requires solving (3), (4), (6), which is not an easy task. Could you also comment on the complexity?
> >
> > That being said, the result achieved seems to be major, yet the presentations would require serious improvements to match the claimed results.

---

> > > ### Author Response · Authors · 2025-08-02
> > > **Thanks a lot for your prompt reply!**
> > >
> > > Thanks a lot for your prompt reply and your further questions!
> > >
> > > ---
> > >
> > > **Re: Solving (3), (4), (6)**
> > >
> > > As described in Step 2 in Figure 1, the problem given by (3), (4), (6) is a **linear constraint satisfiability problem, so it can be solved efficiently via linear programming**. (Namely, it is a linear program without an objective function.) It is well-known that linear programming is tractable — it can be solved by algorithms such as the ellipsoid method in polynomial time. The way we linearize (3) and (4) to obtain the linear constraint satisfiability problem is mentioned at the bottom of Page 6, with details provided in Appendix B due to the page limit.
> > >
> > > ---
> > >
> > > **Re: How did we handle history-dependent policies**
> > >
> > > Let us first point out two misconceptions here, which we think might have led to your confusion:
> > >
> > > - First, **the challenge of optimizing history dependent policies is NOT due to the randomness of POMDPs** because even in full information settings without any randomness, the number of possible histories _still_ grows exponentially. This is because each history trajectory involves not only the states encountered during the play but also the players’ actions. Even when each player has only two possible actions to choose from at each time step, there will be $2^t$ possible combinations in terms of actions played in $t$ time steps. To **explicitly** define a history-dependent policy, one would need to specify what to do (i.e., a joint action distribution) for each of these possible histories. We address this challenge by abandoning explicit representation and resorting to a compact representation (which I will explain further below).
> > >
> > > - Second, **our algorithm does NOT solve POMDPs in their original form** — which is not possible in polynomial time because POMDPs are PSPACE-hard. What we solve is **POMDP under the special condition of hindsight observability**. This condition spares us the pain of dealing with belief states because after each time step both players know all information in the game in the previous time steps; uncertainties exist only temporarily in the current time step. Belief states are the main source of the computational challenges facing POMDPs. Without these challenges, the problem we face becomes purely about dealing with history-dependent strategies, which is the focus of our paper.
> > >
> > > More specifically, our approach consists of the following two parts (corresponding to Sections 3.1 and 3.2, respectively):
> > >
> > > 1. First, we introduce the key concept of **inducible value set — that is, the set of value vectors that can be successfully induced if the principal plays in some way**. In Section 3.1, we showed that these sets can be constructed efficiently via dynamic programming. Once we have the inducible value set of **the starting state at time step 1**,  the best value vector in this set is then, by definition, the best we can achieve in the entire game by using some history-dependent policy. It then remains to find out this policy.
> > >
> > > 2. Now, because of the representational issue we described above, we cannot hope to obtain the explicit representation of this policy; we represent it using an algorithm instead. This is what the algorithm in Figure 2 does. (In other words, you can view the algorithm itself as the policy — a policy given by a Turing Machine.) **If we feed any history $(\sigma, \omega^P, \tilde{\omega}^A)$ as input to the algorithm, the algorithm returns the distribution $\pi(\cdot | \sigma, \omega^P, \tilde{\omega}^A)$ — that is, the action distribution specified by the policy when this given history trajectory $(\sigma, \omega^P, \tilde{\omega}^A)$ is encounterd.** (This is why it is a history-dependent policy.)
> > >
> > > In other words, if we enumerate every possible history, feed them as input to the algorithm in Figure 2, and record the outputs, we will get an explicit representation of the policy. When we actually execute the policy, however, there is no need to precompute this entire representation because only one history trajectory will be realised throughout the play. It then suffices to keep track of only the realised history, and feed it to the algorithm in each time step to obtain the action distribution defined by $\pi$, which is all that the principal needs to know in order to execute the policy.
> > >
> > > ---
> > >
> > > We acknowledge that the description of our algorithm is quite involved, but this is partly also due to the intricate nature of the problem. Perhaps it would be easier to understand the core of the algorithm if we first present the algorithm for a simpler case without information asymmetry. Even in this simpler case, the main challenge of the problem—the exponential growth of history—remains, but the key ideas behind our algorithm would perhaps stand out more clearly.
> > >
> > > Thank you again for your very thoughtful questions! Please feel free to let us know if anything else remains unclear.

---

> > > > ### Comment · Reviewer_o1dQ · 2025-08-02
> > > >
> > > > Thank you for the clarification. I believe that this explanation should be included in the revised version.

---

> > > > > ### Author Response · Authors · 2025-08-03
> > > > >
> > > > > Sure we will! Thanks for your valuable feedback.

---

### Official Review · Reviewer_RCDp · 2025-07-03

**Clarity:** 3
**Significance:** 2
**Originality:** 3
**Rating:** 4
**Confidence:** 3

**Summary:**

The paper analyzes a sequential principal-agent game in which, at every round, the principal observes her private signal, elicits (truthfully or not) the agent’s private signal, and then sends non-binding action recommendation for the agent.  Crucially, it imposes hindsight observability: once the round ends, the entire log—hidden state, both private signals, recommendation, chosen actions, and pay-offs—becomes common knowledge.

With the transition dynamics known, the authors show that an approximate incentive-compatible commitment policy can be computed in polynomial time.  The key device is the inducible value set: a two-dimensional convex polytope that lists every pair of (principal payoff, agent payoff) that remains achievable from a given state-action pair while keeping the agent willing to comply.  A backward dynamic program updates these polytopes, yielding both the optimal value and a constructive policy.

When transition dynamics are unknown, the method is extended with reward-free exploration. After collecting enough trajectories, a modified version of the algorithm yields an approximate incentive complatible policy with regret growing as T^2/3.

**Questions:**

- In settings where only part of the interaction log can be revealed, do you see a principled way to adapt inducible value sets so that tractability is preserved, perhaps with approximation guarantees?
- Do you intend to implement the algorithm on synthetic or real-world benchmarks to test runtime and sample complexity? If such experiments are not planned, what do you see as the primary bottlenecks for implementation?

**Ethical Concerns:**

["NO or VERY MINOR ethics concerns only"]

**Final Justification:**

I’m satisfied with the authors’ rebuttal and maintain my positive rating for the paper.

**Limitations:**

Yes

**Quality:**

3

**Strengths And Weaknesses:**

Strengths
- The paper is well written: definitions, lemmas and algorithms are laid out cleanly, and intuition is provided alongside formal results.
- The paper introduces hindsight observability, a theoretically valuable assumption that renders the model tractable yet still incentive-rich.
- The introduction of inducible value sets, enabling a polynomial-time dynamic-programming solution while embedding incentive constraints, is a solid technical contribution.

Weaknesses
- Despite clean, hindsight observability means that every private signal and action becoming public at the end of each round, which significantly limits practical reach.
- As the author also acknowledges, the method does not scale well to multiple followers, further restricting its scope.
- The paper provides no empirical validation, so the practical runtime and sample-complexity costs remain untested.

---

> ### Author Rebuttal · Authors · 2025-07-30
>
> Thank you for your very insightful and valuable comments and your positive assessment of our work!
>
> ---
>
> Before we address your individual comments/questions below, we’d like to first highlight the following main points:
>
> - The limitations and scalability issue of our approach **are due to computational complexity barriers intrinsic to the problem itself, rather than the deficiency of our algorithm.** We will explain this in detail below.
>
> - More importantly, even though our approach is still subject to these computational barriers, **traditional approaches based on stationary strategies would only be worse:** they are intractable even when there are only a constant number of players and when hindsight observability is assumed.
>
> - The main insight of our results is that **history-dependent strategies—though seemingly unamenable at first glance—become feasible with our approach and, thanks to that, we are able to overcome critical limitations of stationary strategies.** This makes our approach computationally more advantagious than approaches based on stationary strategies, which dominate the current literature.
>
> ---
>
> Regarding your specific questions:
>
> **1. Hindsight observability is inevitable due to intrinsic computational complexity barriers**
>
> The assumption of hindsight observability, even though restricted, is inevitable as the problem would subsume POMDPs without such assumptions. POMDPs are known to be intractable (PSPACE-hard). Hence, unless PSPACE = P (which is widely believed unlikely as this would be even stronger than P = NP), it would not be possible to find any efficient algorithm without making additonal assumptions on the structure of the problem. (Please also see Appendix C in our supplementary material for a detailed discussion about the intractablility without hindsight observability.)
>
> **2. Scaling to multiple followers is NP-hard even in one-shot settings (as implied by [Papadimitriou & Roughgarden, 2008])**
>
> Our solution concept generalizes the correlated equilibrium. Since our goal is to find an **optimal** equilibrium, with $n$ players, the problem is unfortunately known to be NP-hard even for one-shot games [Papadimitriou & Roughgarden, 2008].
>
> **3. Settings where only part of the interaction log is revealed**
>
> In settings where only part of the interaction log is revealed (say, only the state is revealed), our algorithm based on inducible value sets is still applicable and will generate a policy in polynomial time. We do not know if this policy has any approximation guarantee though, **but we can at least be certain that it outperforms the best stationary policy.** Since different players may observe different part of the interaction log under partial observability, to compute optimal policies in such settings requires working with belief states, but the belief state space grows exponentially with the size of the problem instance. **This is exactly also where the intractablity of POMDP stems from, so in general there is no efficient approach to deal with such settings.** It is possible that under some structural assumptions weaker than hindsight observability, the problem remains tractable. To identify such assumptions (and whether they exist at all) require further investigation, which we think is beyond the scope of this present work.
>
> **4. Re: Experiments**
>
> Since our aim is to establish the theoretical foundation of the model, especially the tractability of history-dependent strategies, we didn’t conduct any experiments in the current work. We are indeed interested in evaluating our approach empirically, and we do not see any sever bottleneck for implementing the algorithm. Scalability may be one issue if one wants to implement the algorithm for many agents/followers. In this case, heuristic algorithms would be a more practical option, as is the case with many other NP-hard problems. We believe that our results remain constructive in this domain and inform the design of heuristic algorithms about the importance and advantage of history-dependent strategies.
>
> ---
>
> **Reference:**
>
> - Christos H Papadimitriou and Tim Roughgarden. Computing equilibria in multi-player games. SODA ‘05.
>
> ---
>
> Please let us know if you have any further questions. We would be more than glad to provide additional clarifications.

---

### Note · Authors · 2025-08-12

We sincerely thank all the reviewers for taking time to assess our work and for the very constructive discussions. We believe that all the reviewers are satisfied with our responses and think positively about our work. For the AC's convenience, we wish to highlight the following two clarifications that were discussed in detail:

- **Hindsight Observability.** The condition of hindsight observability is proposed due to the PSPACE-hard barrier of POMDPs, which our model would subsume if hindsight observability does not hold. We identify hindsight observability as a natural and broadly construed condition under which the computational barrier can be circumvented to allow for efficient (polynomial-time) algorithms. The condition encompasses models of wide interest, including repeated games, standard Markov games, and sequential information design, while we also provide real-world examples in trading and R&D consortium that fall under this condition (please see our response to Reviewer o9Qq).

- **Positioning relative to prior and concurrent work.** Our work improves upon prior work by enabling history-dependence in optimal policy design. This capability, in contrast to widely employed stationary strategies, not only improves the policy designer’s utility but also turns out to be computationally more advantageous, thanks to our efficient algorithm. Our results strictly outperform concurrent work by achieving _exact_ IC (incentive compatibility) on a richer model that allows for two-sided information asymmetry and two-way information exchange between the principal and the agent, while we have also investigated the learning version of the problem and achieved efficient no-regret learning.

We hope that this final remark will be helpful for your assessment and thank you for your time and expertise.

---

### Decision · Program_Chairs · 2025-09-17

**Decision:**

Accept (poster)

**Comment:**

The paper shows that optimal and exactly IC solutions can be computed efficiently in two-player stochastic games with asymmetric information by extending a recent idea of using value polytopes.  The reviewers unanimously appreciate the novelty, importance, and technical depth of the results.  There were a number of questions and concerns, most of which were addressed during extensive author-reviewer discussion.  Altogether we believe the paper would be a valuable addition to the program.  We encourage the authors to further improve the paper based on the constructive comments.